# Testing the ion-current model for flagellar length sensing and IFT regulation

**Hiroaki Ishikawa[1], Jeremy Moore[2†], Dennis R Diener[3], Markus Delling[4], Wallace F Marshall[1]\***

[1]Department of Biochemistry and Biophysics, University of California, San Francisco, San Francisco, United States; [2]Kenyon College, Gambier, and Summer Research Training Program at University of California San Francisco, San Francisco, United States; [3]Max Planck Institute of Molecular Cell Biology and Genetics, Dresden, Germany; [4]Department of Physiology, University of California, San Francisco, San Francisco, United States

**\*For correspondence:**
wallace.marshall@ucsf.edu

**Present address:** [†]Department of Molecular Cellular and Developmental Biology, Yale University, New Heaven, United States

**Competing interest:** The authors declare that no competing interests exist.

**Abstract** Eukaryotic cilia and flagella are microtubule-based organelles whose relatively simple shape makes them ideal for investigating the fundamental question of organelle size regulation. Most of the flagellar materials are transported from the cell body via an active transport process called intraflagellar transport (IFT). The rate of IFT entry into flagella, known as IFT injection, has been shown to negatively correlate with flagellar length. However, it remains unknown how the cell measures the length of its flagella and controls IFT injection. One of the most-discussed theoretical models for length sensing to control IFT is the ion-current model, which posits that there is a uniform distribution of $Ca^{2+}$ channels along the flagellum and that the $Ca^{2+}$ current from the flagellum into the cell body increases linearly with flagellar length. In this model, the cell uses the $Ca^{2+}$ current to negatively regulate IFT injection. The recent discovery that IFT entry into flagella is regulated by the phosphorylation of kinesin through a calcium-dependent protein kinase has provided further impetus for the ion-current model. To test this model, we measured and manipulated the levels of $Ca^{2+}$ inside of *Chlamydomonas* flagella and quantified IFT injection. Although the concentration of $Ca^{2+}$ inside of flagella was weakly correlated with the length of flagella, we found that IFT injection was reduced in calcium-deficient flagella, rather than increased as the model predicted, and that variation in IFT injection was uncorrelated with the occurrence of flagellar $Ca^{2+}$ spikes. Thus, $Ca^{2+}$ does not appear to function as a negative regulator of IFT injection, hence it cannot form the basis of a stable length control system.

## Editor's evaluation

This paper is valuable and of interest to scientists studying primary cilia/flagellar formation and regulation. It addresses how ciliary/flagellar length is controlled and whether calcium negatively regulates Intraflagellar transport (IFT) injection. The study convincingly demonstrates that calcium influx correlates with flagellar length, but calcium does not appear to work as a negative regulator of IFT injection, which challenges a previous model. The models and methods are generally sound.

## Introduction

The mechanism by which cells control the size of organelles is a fundamental question in cell biology. Cells control the size of their organelles to efficiently use their energy and space (*Chan and Marshall, 2012*). However, it is mostly unknown how cells regulate organelle size, in part because the complexity of many organelles makes their size control mechanisms difficult to study. Eukaryotic cilia and flagella

(we use these terms interchangeably) are an ideal model system to study size regulation because they consist of a simple, hair-like structure that can change in length while maintaining a constant diameter. Cilia and flagella are microtubule-based organelles that protrude from the cell body and are important for sensing extracellular signals and producing fluid flows and cell locomotion (*Ishikawa and Marshall, 2011*; *Reiter and Leroux, 2017*). It is known that the length of cilia and flagella is cell type specific. For example, the biflagellate green alga *Chlamydomonas reinhardtii* adjusts the length of both flagella to a length suitable for swimming (*Bauer et al., 2021*; *Bottier et al., 2019*). If the flagella are too long or too short, the cells cannot swim efficiently. When flagella are removed from a cell, new flagella regenerate to the pre-severing length in around 90min (*Rosenbaum et al., 1969*). Flagellar growth slows before they become full-length. This decelerating growth rate suggests that there is some mechanism that regulates the growth rate as a function of length.

Assembly and maintenance of flagella are dependent on an active transport system, intraflagellar transport (IFT). Because no protein synthesis occurs in the flagellum, new materials must be brought into the flagellum and carried to the site of assembly at the flagellar tip. IFT mediates this transport. IFT is carried out by huge protein complexes called trains, which have an elongate shape and move on the flagellar axoneme back and forth like a train (*Jordan et al., 2018*; *Pigino et al., 2009*). IFT trains assemble around the basal body and then enter the flagellum in a process known as IFT injection, after which they are carried to the flagellar tip by a heterotrimeric kinesin-2 motor and returned to the cell body by cytoplasmic dynein 2 (*Ishikawa and Marshall, 2011*; *Rosenbaum and Witman, 2002*; *van den Hoek et al., 2022*). It is known that the rate of IFT injection is negatively correlated with the length of flagella during flagellar regeneration (*Engel et al., 2009*; *Ludington et al., 2013*), but it is unknown how cells regulate the rate of IFT injection. To explain this negative correlation, several theoretical models have been proposed (*Chan and Marshall, 2012*; *Ludington et al., 2015*; *Rosenbaum, 2003*).

One well-known model for the regulation of IFT injection is the ion-current model (also known as the ciliary-current model). The ion-current model was first proposed by *Rosenbaum, 2003* based on a previous electrophysiological study showing that voltage-gated calcium channels were present in flagella at constant density (*Beck and Uhl, 1994*), such that the number of voltage-gated calcium channels in the flagellum increases as the flagellum elongates. Changes in flagellar calcium concentrations, due to changes in the number and activity of calcium channels in the elongating flagella, were proposed to affect calcium-dependent signaling pathways to regulate the rate of IFT injection or microtubule assembly (*Rosenbaum, 2003*). In fact, changes in external $Ca^{2+}$ are known to change flagellar length in *Chlamydomonas* (*Lefebvre et al., 1978*; *Quader et al., 1978*; *Tuxhorn et al., 1998*). The calcium ion-current model was first proposed based on physiological measurements, but at the time, the molecular identity of the voltage-gated calcium channel in the flagellar membrane was not known. More recently, several calcium channel proteins have been shown to localize on the *Chlamydomonas* flagellum (*Fujiu et al., 2011*; *Fujiu et al., 2009*; *Huang et al., 2007*; *Pazour et al., 2005*). CAV2 is a subunit of a voltage-gated calcium channel which resides in the flagellar membrane (*Fujiu et al., 2009*). The *ppr2* mutant has a mutation in the CAV2 gene and is defective in the photo-phobic response (*Fujiu et al., 2009*; *Matsuda et al., 1998*). TRP channels are also found in *Chlamydomonas* flagella (*Fujiu et al., 2011*; *Huang et al., 2007*).

In the original ion-current model, it was not specified how $Ca^{2+}$ contributes to flagellar length regulation, but a potential molecular mechanism has been proposed by Pan and colleagues (*Liang et al., 2014*; *Liang et al., 2018*). They found that a subunit of heterotrimeric kinesin that drives IFT anterograde motion, FLA8/KIF3B, is phosphorylated by a calcium-dependent protein kinase (CDPK1), and this phosphorylation negatively regulates IFT injection into flagella. Because this work provides a potential missing link in understanding the ion-current model, calcium-mediated regulation of IFT entry has increasingly been viewed as a likely means of regulating IFT in response to flagellar length (*Engelke et al., 2019*; *Jiang et al., 2019*; *Kumari and Ray, 2022*; *Lechtreck et al., 2017*; *Liang et al., 2018*). There is thus a pressing need to test whether this model does in fact account for IFT regulation as a function of length. The ion-current model hinges on two critical assumptions (*Figure 1A*), both of which are experimentally testable. First, the amount of flagellar $Ca^{2+}$ needs to be an increasing function of the flagellar length. This assumption would allow $Ca^{2+}$ to serve as a length indicator within the ion-current model. Second, for the ion-current model to work, cellular $Ca^{2+}$ must negatively regulate IFT injection or assembly of the axoneme. This is because if the amount of flagellar $Ca^{2+}$ increases with

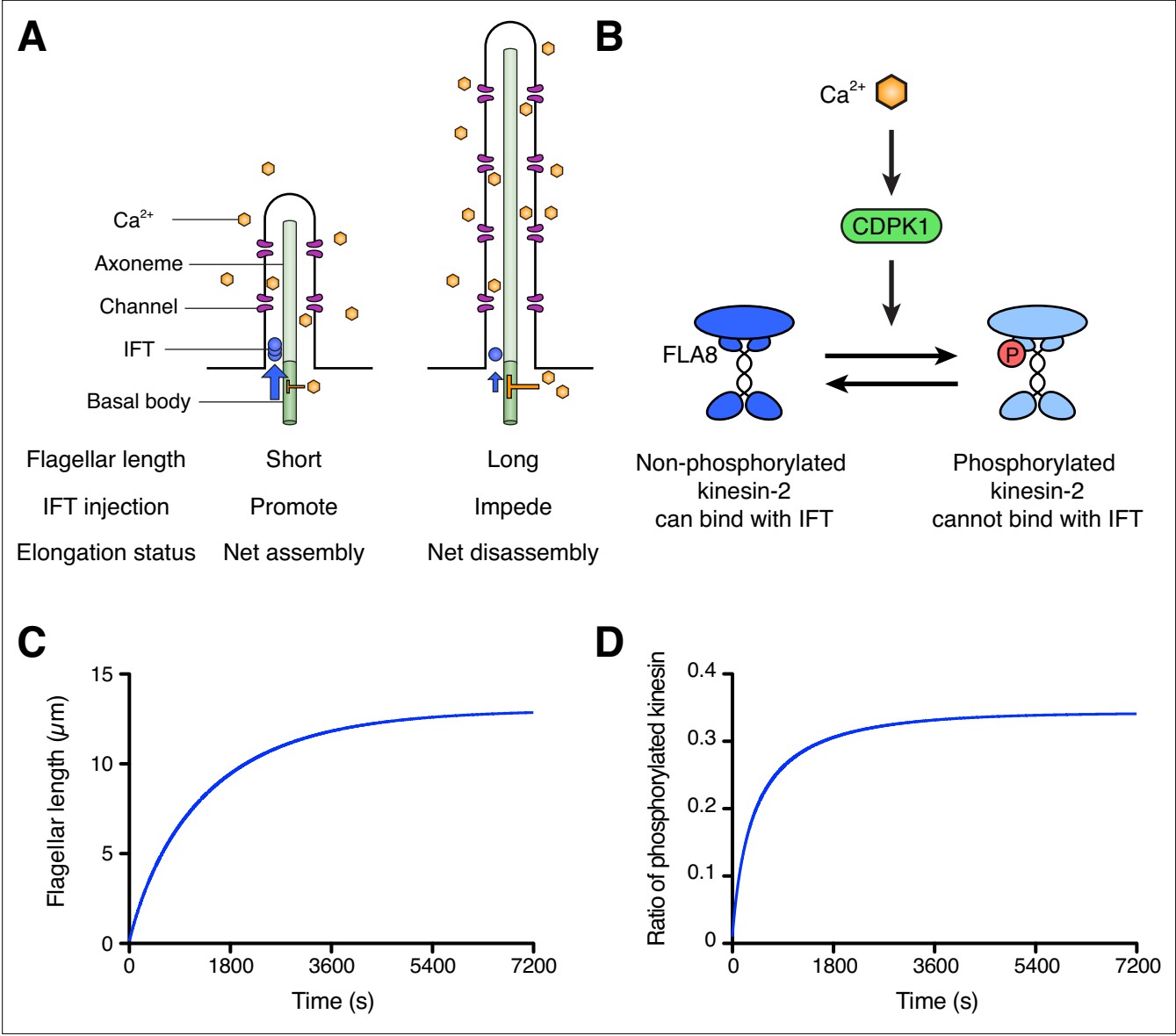

**Figure 1.** Schematic diagram and modeling of the ion-current model. (**A**) The ion-current model assumes the ion channels are uniformly distributed along the length of the flagellum, and $Ca^{2+}$ ions entering the flagellum are proportional to the flagellar length. Flagellar $Ca^{2+}$ is assumed to negatively regulate IFT injection into flagella to control the flagellar length. The longer flagellum can intake more ions, and these ions inhibit IFT injection such that the further assembly of flagella is suppressed. (**B**) The schema of kinesin-2 inactivation by $Ca^{2+}$ and CDPK1 based on *Liang et al., 2014*. In a $Ca^{2+}$-dependent manner, CDPK1 phosphorylates FLA8, a subunit of heterotrimeric kinesin-2. Phosphorylated kinesin-2 loses its IFT protein binding activity. (**C, D**) Simulated result of the ion-current model based on kinesin-2 phosphorylation by CDPK1 (*Liang et al., 2014*), as detailed in Materials and methods. Flagellar length (**C**) and the ratio of phosphorylated kinesin (**D**) were plotted against time.

the flagellar length, it must work as a negative regulator to control the length of flagella in order to have a stable length control system with negative feedback. If $Ca^{2+}$ positively regulated IFT injection, it could not be part of a stable feedback control loop and would instead produce positive feedback.

Here, we tested the two underlying assumptions of the ion-current model by quantitatively measuring the $Ca^{2+}$ concentration inside of *Chlamydomonas* flagella using a genetically encoded calcium biosensor, together with IFT injection, in wild-type and calcium-channel deficient flagella. The amount of $Ca^{2+}$ inside of flagella was indeed correlated with the length of flagella, as the model requires. However, IFT injection was reduced in calcium-deficient flagella, rather than increased as the model predicted. Moreover, $CaCl_2$-treated cells increased the amount of flagellar $Ca^{2+}$ and IFT injection. We then examined fluctuations of IFT at the time scale of individual injection events and

compared these fluctuations to fluctuations in flagellar $Ca^{2+}$ and found no correlation, suggesting that flagellar $Ca^{2+}$ does not influence IFT injection at a short timescale. Thus, flagellar $Ca^{2+}$ apparently does not work as a negative regulator of IFT injection and therefore could not form the basis of a stable length control system. These observations are thus inconsistent with the ion-current model.

## Results

### Model for the ion-current model using kinesin phosphorylation

Before testing the calcium-based model experimentally, we first asked whether it could provide a stable length control mechanism, even in theory. We constructed a simplified model for flagellar length dynamics assuming that calcium entry is proportional to the flagellar length and that kinesin-2 activity is regulated by phosphorylation mediated by CDPK1 as previously reported (*Liang et al., 2014*) and summarized in *Figure 1B*. Details of this model, which integrates IFT-mediated transport, $Ca^{2+}$ influx, and CDPK activity modulating IFT entry, are provided in Materials and Methods. As shown in *Figure 1C*, this model is able to produce a stable length control system, such that when flagella are removed they grow back to the correct length with decelerating kinetics that resemble those seen in actual flagella. *Figure 1D* further shows that the relative level of phosphorylation of kinesin changes during regeneration, smoothly increasing as the length increases, consistent with the experimental observations reported by *Liang et al., 2014*. We note that this is a highly simplified model, and detailed numerical comparison to experiments is not useful since a number of parameters have unknown values. Nevertheless, this model serves to confirm the intuition that $Ca^{2+}$ influx proportional to flagellar length, combined with CDPK-mediated regulation of IFT injection could, at least in principle, serve as a length control mechanism.

### Quantitative flagellar $Ca^{2+}$ detection using GCaMP

The first assumption of the ion-current model that we explore is the requirement that the $Ca^{2+}$ amount in the flagellum should be an increasing function of flagellar length. To measure the amount of $Ca^{2+}$ in *Chlamydomonas* flagella, we constructed *Chlamydomonas* strains that express GCaMP, a genetically encoded calcium indicator (*Nakai et al., 2001*), anchored in its flagella. GCaMP was fused to the C-terminus of dynein regulatory complex 4 (DRC4), an axonemal protein (*Rupp and Porter, 2003*), and DRC4-GCaMP was expressed in the *pf2-4* mutant cells, a loss-of-function mutant of DRC4. DRC4-GCaMP was localized along the length of flagella (*Figure 2A*) and rescued the defective flagellar motility phenotype of the *pf2-4* mutant. DRC4 is a component of the Nexin-Dynein Regulatory Complex, which is a part of the 96 nm axonemal repeat, periodical structures on the axoneme (*Heuser et al., 2009*; *Rupp and Porter, 2003*). Therefore, the amount of DRC4-GCaMP is constant per unit length, and the total quantity of DRC4-GCaMP associated with axonemes is determined solely by the length of the flagella. We previously showed that the amount of axonemal proteins, such as FAP20 and RSP3, is constant per unit length using GFP (*Ishikawa et al., 2022*). This result means that our system using DRC4-GCaMP should be able to quantitatively detect flagellar $Ca^{2+}$.

To confirm that axonemal DRC4-GCaMP produces a signal that depends quantitatively on free $Ca^{2+}$, we measured the GCaMP intensity using isolated DRC4-GCaMP axonemes. We isolated flagella from DRC4-GCaMP (or DRC4-mCherry-GCaMP) cells and removed the membrane from flagella so that $Ca^{2+}$ can access the axoneme-associated DRC4-GCaMP (or DRC4-mCherry-GCaMP). Isolated axonemes were observed in various concentrations of free $Ca^{2+}$ solutions (*Figure 2B*) and the intensity was quantified (*Figure 2C*). GCaMP fluorescence showed a sigmoidal curve with respect to free $Ca^{2+}$ concentration, with an approximately linear response in the range of free $Ca^{2+}$ from 0.05 to 1 μM. *Figure 2C* provides a calibration curve that lets us quantitatively measure $Ca^{2+}$ in *Chlamydomonas* flagella based on DRC4-GCaMP fluorescence.

### The $Ca^{2+}$ influx into the flagellum is correlated with flagellar length

To check whether the amount of $Ca^{2+}$ influx into the flagellum is increased with increasing flagellar length, we observed DRC4-GCaMP in flagella of living cells using total internal reflection fluorescence (TIRF) microscopy during flagellar regeneration. *Chlamydomonas* flagella were amputated with the pH shock method (see Materials and methods), allowing us to measure $Ca^{2+}$ influx as a function of flagellar length as the flagella regrow. We observed intensity fluctuation of GCaMP in live cells,

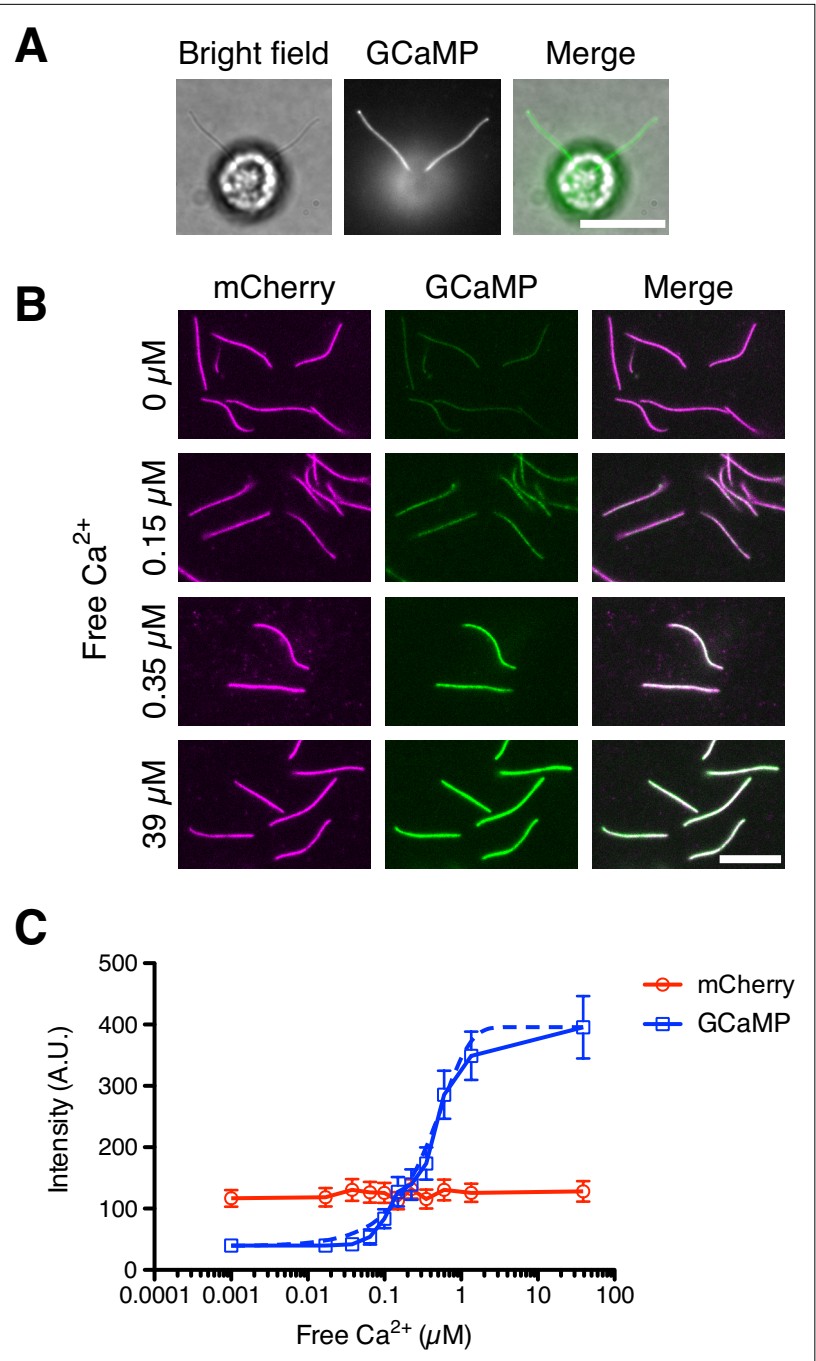

**Figure 2.** DRC4-GCaMP quantitatively detects free Ca$^{2+}$. (**A**) Bright-field and fluorescent images of DRC4-GCaMP cells. DRC4-GCaMP localizes to the entire length of flagella. Scale bar: 10 µm. (**B**) Fluorescent images of isolated DRC4-mCherry-GCaMP axonemes. Axonemes were isolated from DRC4-mCherry-GCaMP cells and were treated with various concentration of free Ca$^{2+}$. GCaMP intensity increased as free Ca$^{2+}$ concentration increased. Scale bar: 10 µm. (**C**) Semilogarithmic plot of GCaMP and mCherry intensities. GCaMP and mCherry intensities were measured and plotted with mean ± SD. Thirteen axonemes were analyzed for each Ca$^{2+}$ concentration. The blue dashed line shows the equation of a sigmoidal curve which is calculated from the data.

The online version of this article includes the following source data for figure 2:

**Source data 1.** Raw data of DRC4-mCherry-GCaMP intensity.

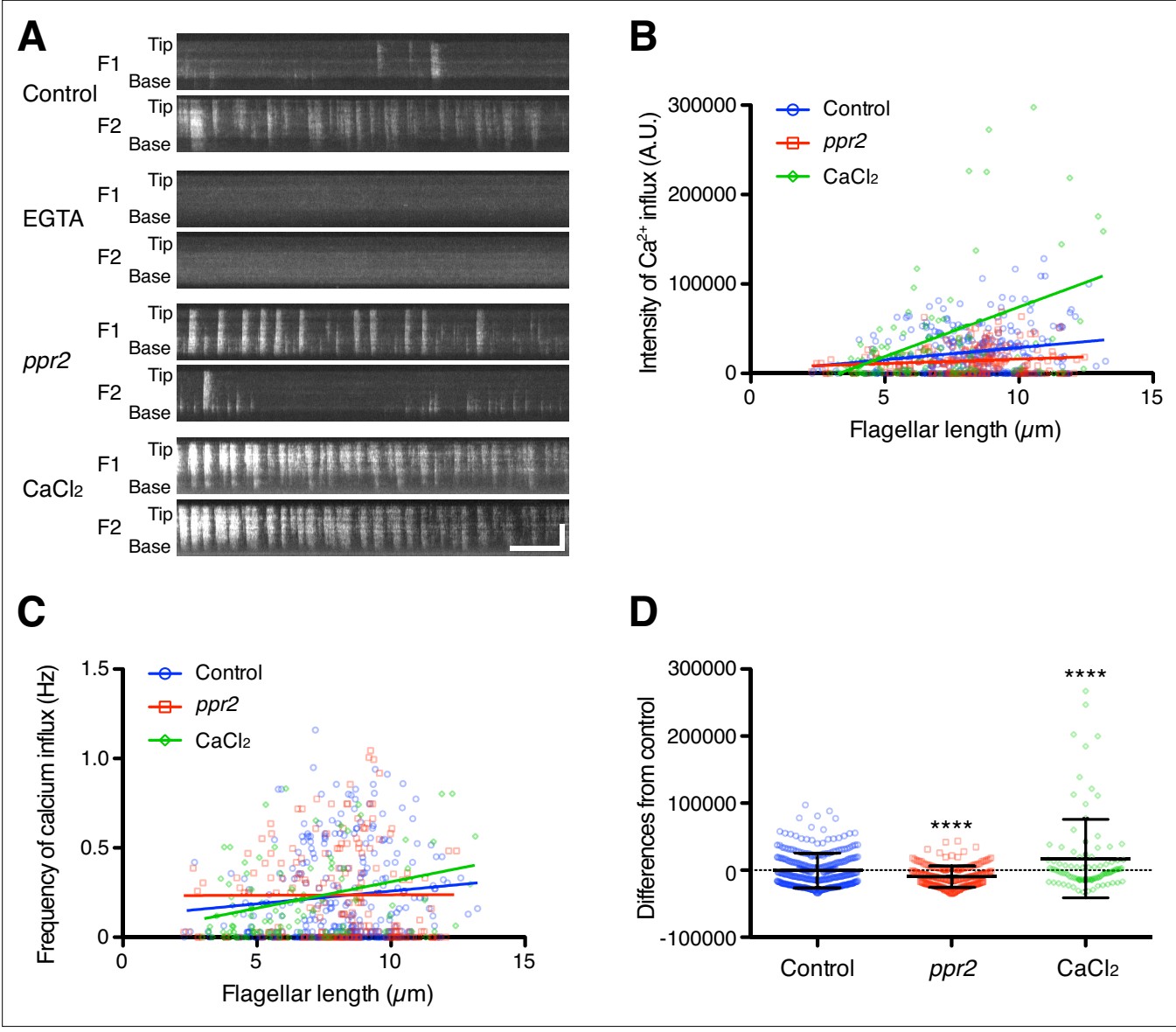

**Figure 3.** Quantification of Ca2 +influx as a function of flagellar length. (**A**) Representative DRC4-GCaMP kymographs of *Chlamydomonas* flagella in control (*pf2* DRC4-GCaMP), 1 mM EGTA-treated (*pf2* DRC4-GCaMP), *ppr2* mutant (*pf2 ppr2* DRC4-GCaMP), and 1 mM CaCl$_2$-treated (*pf2* DRC4-GCaMP) cells. These kymographs were assembled from *Videos 1–4*. Horizontal bar: 5 s; vertical bar: 5 µm. (**B**) The intensity of Ca$^{2+}$ influx into flagella was calculated from kymographs and plotted against flagellar length. Different lengths of flagella were obtained by imaging flagella during regeneration. Control (blue circles, n=272, Pearson correlation coefficient $\rho$ =0.21, and coefficient of determination r$^2$=0.04), the *ppr2* mutant (red squares, n=182, $\rho$ =0.14, and r$^2$=0.02), and 1 mM CaCl$_2$-treated cells (green diamonds, n=96, $\rho$ =0.46, and r$^2$=0.21). (**C**) The frequency of Ca$^{2+}$ influx was plotted against flagellar length. No obvious correlation was detected in either the control (blue circles, $\rho$ =0.11, and r$^2$=0.01) or the *ppr2* mutant (red squares, $\rho$ =0.004, and r$^2$=1.85 × 10$^{-5}$). However, the frequency of Ca$^{2+}$ influx in 1 mM CaCl$_2$ treated cells was correlated with flagellar length (green diamonds, $\rho$ =0.33, and r$^2$=0.11). (**D**) The mean differences of Ca$^{2+}$ influx intensity from the control regression line. Data were plotted as scatter dot plots with mean ± SD. Statistical significance was determined by an unpaired two-tailed t test against the control (**** p<0.0001).

The online version of this article includes the following source data and figure supplement(s) for figure 3:

**Source data 1.** Raw data of flagellar length and DRC4-GCaMP intensity.

**Figure supplement 1.** Example DRC4-GCaMP kymographs of *Chlamydomonas* flagella (control, *pf2* DRC4-GCaMP).

**Figure supplement 2.** Individual Ca$^{2+}$ influx events are not independent.

indicating the presence of Ca²⁺ influx (*Figure 3A* and *Video 1*). This intensity fluctuation was not observed when 1 mM EGTA was added to the media (~150 nM of free Ca²⁺, which is the same level as the cytosolic Ca²⁺ concentration; *Figure 3A* and *Video 2*). This result indicated that most of the Ca²⁺ influx came from outside of the cell.

TIRF movies of DRC4-GCaMP were converted into kymographs to measure the calcium amount in the flagellum (*Figure 3A* and *Videos 1–4*). The amount of flagellar Ca²⁺ influx was significantly correlated with the flagellar length but with a broad distribution of values around the trend line (*Figure 3B*, Pearson correlation coefficient $\rho$ =0.33, and coefficient of determination $r^2$=0.11). Longer flagella generally allowed more Ca²⁺ influx but also had greater variability. This variability is mainly caused by the fact that the Ca²⁺ influx does not occur consistently. The frequency of flagellar Ca²⁺ influx events (spikes) was not correlated with the flagellar length (*Figure 3C*, $\rho$ =0.14, and $r^2$=0.02). The pattern of Ca²⁺ influx events varied from flagellum to flagellum and from time to time and did not match a Poisson distribution (*Figure 3—figure supplements 1 and 2*). Consistent with previous reports, the Ca²⁺ influx into the flagellum was compartmentalized, such that there was no correlation in Ca²⁺ influx events between the two flagella from the same cell (*Figure 3A*, *Figure 3—figure supplement 1*, and *Video 1*; *Collingridge et al., 2013*).

To reduce the Ca²⁺ influx into the flagellum, we used a calcium channel mutant strain, *ppr2* (*Fujiu et al., 2009*; *Matsuda et al., 1998*). The *ppr2* mutant has an insertion in the CAV2 gene, which encodes a voltage-gated calcium channel localized to flagella (*Fujiu et al., 2009*). In the *ppr2* mutant, we still observed Ca²⁺ influx events (*Figure 3A* and *Video 3*), but the amount of Ca²⁺ influx was no longer correlated with the flagellar length (*Figure 3B*, $\rho$ =0.09, and $r^2$=0.02) and the magnitude of the influx was smaller than control on the average (*Figure 3D*). Interestingly, the *ppr2* mutant reduced Ca²⁺ influx most strongly at the distal region of flagella (*Figure 3A*). This reduction is presumably caused by the loss of CAV2 protein, which localizes in the distal part of the flagellum (*Fujiu et al., 2009*). Conversely, we tried to increase the amount of flagellar Ca²⁺ by raising the concentration of extracellular Ca²⁺. When 1 mM CaCl₂ was added to the media, we confirmed that the amount of flagellar Ca²⁺ was increased (*Figure 3* and *Video 4*). Thus, we could regulate the amount of flagellar Ca²⁺ by using the flagellar Ca²⁺ channel mutant cells or changing the external Ca²⁺ concentration.

## IFT injection was reduced in calcium-deficient flagella

As discussed above, in order for the ion-current model to work, Ca²⁺ would need to be a negative regulator of IFT injection. Any reduction in Ca²⁺ influx should thus lead to increased IFT injection. Because the amount of Ca²⁺ influx into flagella was reduced in the *ppr2* mutant (*Figure 3B and D*), IFT injection in the *ppr2* mutant should be increased compared to control cells if the ion-current model is correct. To test this prediction, we quantified the IFT injection using kinesin-associated protein (KAP), which is a subunit of the heteromeric kinesin-2 motor (IFT kinesin), tagged with green fluorescent protein (GFP). We focused on the kinesin-2 motor because this is the component of IFT that is apparently regulated by Ca²⁺ (*Liang et al., 2018*; *Liang et al., 2014*). KAP-GFP was expressed in the *fla3* mutant, which has a point mutation in the FLA3 gene (encodes KAP), and rescued the phenotype of the *fla3* mutant (*Mueller et al., 2005*). This *fla3* KAP-GFP strain was also used in previous studies for measuring IFT injection intensity (*Engel et al., 2009*; *Ishikawa and Marshall, 2017*; *Ludington et al., 2013*). Movies of KAP-GFP were converted into kymographs to measure IFT injection (*Figure 4A* and *Videos 5–8*). Similar to previous reports

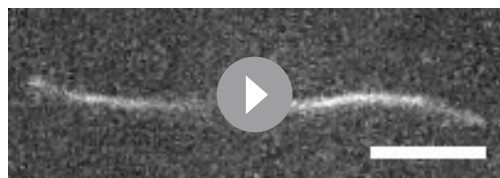

**Video 1.** Movie of DRC4-GCaMP in control (*pf2* DRC4-GCaMP) cell. This movie was taken at 20 frames per second (fps) and plays in real time. Scale bar 5 µm.
https://elifesciences.org/articles/82901/figures#video1

**Video 2.** Movie of DRC4-GCaMP in 1 mM EGTA treated (*pf2* DRC4-GCaMP) cell. This movie was taken at 20 fps and plays in real time. Scale bar 5 µm.
https://elifesciences.org/articles/82901/figures#video2

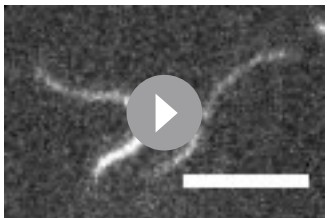

**Video 3.** Movie of DRC4-GCaMP in the *ppr2* mutant (*pf2 ppr2* DRC4-GCaMP) cell. This movie was taken at 20 fps and plays in real time. Scale bar 5 μm.
https://elifesciences.org/articles/82901/figures#video3

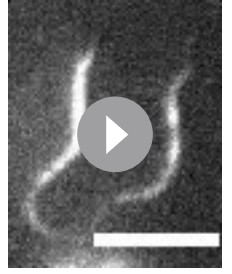

**Video 4.** Movie of DRC4-GCaMP in 1 mM $CaCl_2$ treated (*pf2* DRC4-GCaMP) cell. This movie was taken at 20 fps and plays in real time. Scale bar 5 μm.
https://elifesciences.org/articles/82901/figures#video4

(*Ishikawa and Marshall, 2017*; *Ludington et al., 2013*), the injection intensity of KAP-GFP decreases as flagellar length increases in control cells (*Figure 4B*). In the *ppr2* mutant, IFT injection was also negatively correlated with the flagellar length but was reduced in short flagella compared to control cells (*Figure 4B and C*). This is the opposite of the prediction of the ion-current model. To confirm that flagellar $Ca^{2+}$ is related to this IFT reduction phenotype, we treated KAP-GFP cells with 1 mM EGTA and measured the intensity of IFT injection. EGTA treated cells showed reduced, rather than increased, IFT injection, similar to that seen in *ppr2* mutant cells (*Figure 4B and C*). In contrast, the KAP-GFP cells treated with 1 mM $CaCl_2$ increased the intensity of IFT injection (*Figure 4B and C*). Because reduction in $Ca^{2+}$ caused a decrease, rather than an increase in IFT injection and vice versa, these results suggested that flagellar $Ca^{2+}$ does not work as a negative regulator of IFT injection. Our observations thus do not support the ion-current model.

## Effect of $Ca^{2+}$ on flagellar regeneration and disassembly

The preceding results were based on quantifying KAP entry into flagella. As an alternative indicator of IFT based on IFT function, we note that IFT is thought to drive flagellar elongation, such that increased IFT injection should lead to faster flagellar growth, while decreased IFT injection should lead to slower growth. Flagellar growth can thus be used as an indicator of functionally altered IFT injection. To confirm our result that flagellar $Ca^{2+}$ does not inhibit IFT injection as predicted by the ion-current model, we performed flagellar regeneration assays while altering $Ca^{2+}$ concentrations. *Chlamydomonas* cells were treated with low-pH media to remove their flagella and cultured with TAP media with or without 1 mM EGTA (free $Ca^{2+}$ in the two types of media is 149.3 nM and 170 μM, respectively). A total of 150 nM free $Ca^{2+}$ is comparable to the cytosolic $Ca^{2+}$ concentration but was enough to eliminate detectable flagellar $Ca^{2+}$ influx (*Figure 3A* and *Video 2*). EGTA-treated cells showed slower flagellar regeneration than control cells but could eventually reach the original length of flagella (*Figure 5A*, orange triangles). This observation is consistent with previous studies (*Liang and Pan, 2013*) but is contrary to the prediction of the ion-current model, in which reduction of calcium should lead to faster regeneration due to increased IFT injection.

These findings suggested that at least a cytoplasmic concentration of $Ca^{2+}$ is enough for flagellar regeneration, and further $Ca^{2+}$ influx into the flagellum is not essential. Interestingly, the *ppr2* mutant cells showed slower flagellar regeneration and shorter final length than 1mM EGTA treated cells even though the *ppr2* mutants still have some $Ca^{2+}$ influx events (*Figure 5A*, red square). In any case, $Ca^{2+}$ influx through CAV2 is apparently not necessary for assembly of flagella, and CAV2 might have another role for flagellar assembly or maintenance. To investigate whether extracellular $Ca^{2+}$ can promote flagellar assembly, we added 1mM $CaCl_2$ to TAP media after pH shock. Flagellar regeneration kinetics in $CaCl_2$ added media did not change relative to normal TAP media (*Figure 5A*, green diamonds). The extra amount of extracellular $Ca^{2+}$ does not help assembly of flagella. Because the IFT injection was increased in the $CaCl_2$ treated cells, without causing an increase in flagellar assembly, these IFT trains might carry less cargo, such as tubulin. It has been reported that tubulin loading onto IFT trains is regulated by flagellar length (*Craft et al., 2015*).

While flagellar $Ca^{2+}$ does not seem to affect either IFT injection or flagellar assembly in the way expected from the ion-current model, an alternative length control model involving $Ca^{2+}$ could be that flagellar $Ca^{2+}$ might promote flagellar disassembly, such that longer flagella would experience

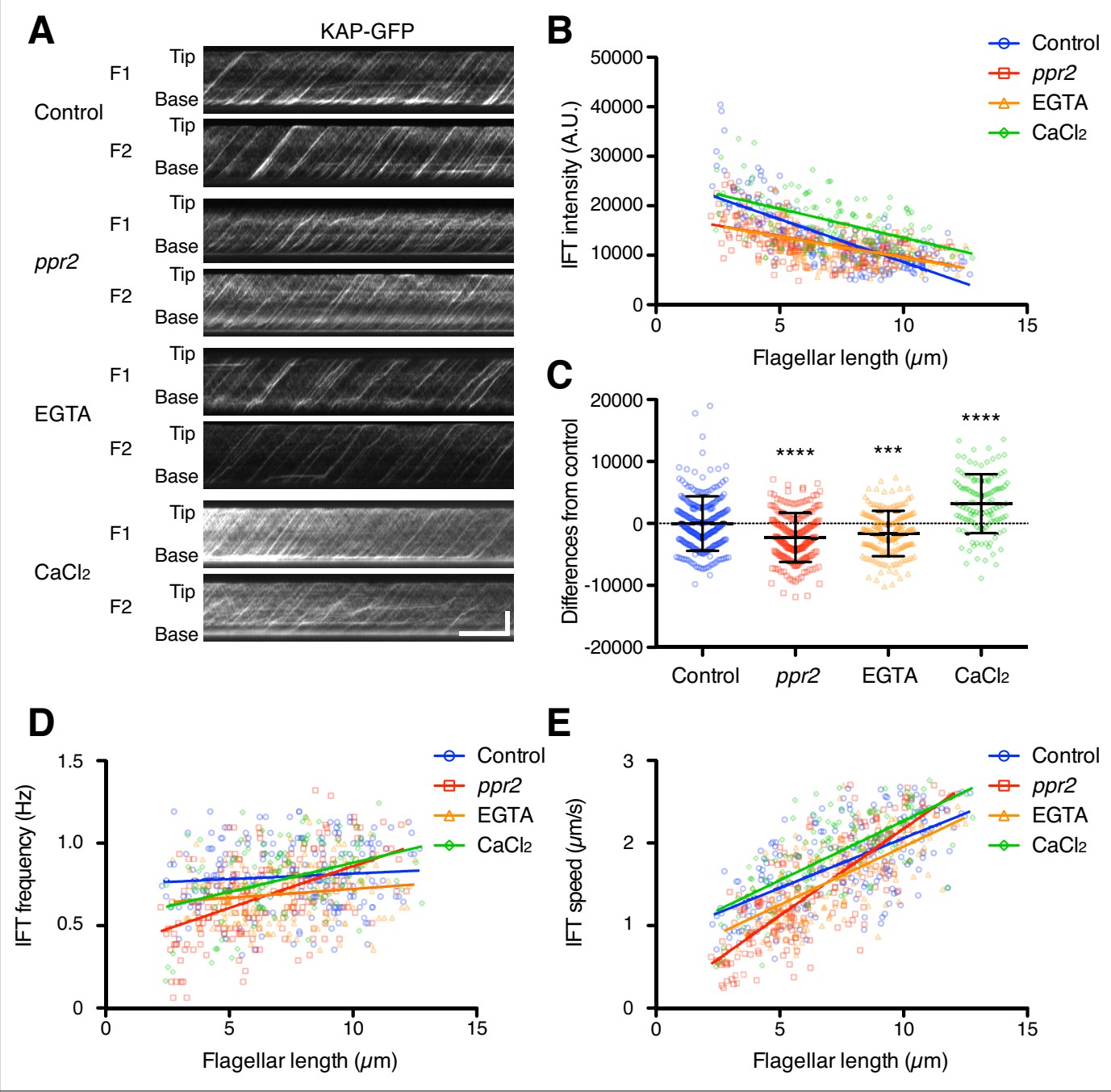

**Figure 4.** Quantification of IFT injection as a function of flagellar length. (**A**) Representative KAP-GFP kymographs of *Chlamydomonas* flagella in control (*fla3* KAP-GFP), *ppr2* mutant (*fla3 ppr2* KAP-GFP), 1 mM EGTA-treated (*fla3* KAP-GFP), and 1 mM CaCl₂-treated (*fla3* KAP-GFP) cells. These kymographs were assembled from *Videos 5–8*. Horizontal bar: 5 s; vertical bar: 5 μm. (**B**) The mean KAP-GFP intensity of each flagellum was calculated from kymographs and plotted against flagellar length. Control (blue circles, n=220, $\rho$ =–0.72, and $r^2$=0.51), *ppr2* (red squares, n=192, $\rho$ =–0.52, and $r^2$=0.27), EGTA (orange triangles, n=162, $\rho$ =–0.53, and $r^2$=0.28), and CaCl₂ (green diamonds, n=116, $\rho$ =–0.52, and $r^2$=0.27). (**C**) The mean difference of IFT intensity from the control regression line. Data were plotted as scatter dot plot with mean ± SD. Statistical significance was determined by an unpaired two-tailed t test against the control (*** $p<0.001$; **** $p<0.0001$). (**D**) The frequency of KAP-GFP was plotted against flagellar length. Control (blue circles, $\rho$ =0.17, and $r^2$=0.03), *ppr2* (red squares, $\rho$ =0.51, and $r^2$=0.26), EGTA (orange triangles, $\rho$ =0.14, and $r^2$=0.02), and CaCl₂ (green diamonds, $\rho$ =0.38, and $r^2$=0.15). (**E**) The velocity of KAP-GFP was plotted against flagellar length. Control (blue circles, $\rho$ =0.64, and $r^2$=0.41), *ppr2* (red squares, $\rho$ =0.81, and $r^2$=0.64), EGTA (orange triangles, $\rho$ =0.69, and $r^2$=0.47), and CaCl₂ (green diamonds, $\rho$ =0.69, and $r^2$=0.48).

The online version of this article includes the following source data for figure 4:

**Source data 1.** Raw data of flagellar length and KAP-GFP intensity.

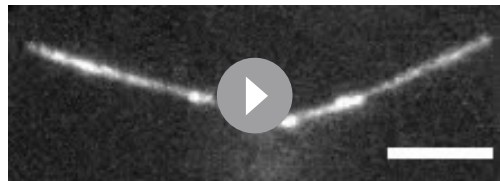

**Video 5.** Movie of KAP-GFP in control (*fla3* KAP-GFP) cell. This movie was taken at 20 fps and plays in real time. Scale bar 5 µm.
https://elifesciences.org/articles/82901/figures#video5

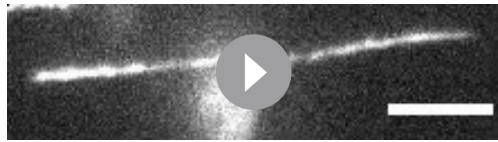

**Video 6.** Movie of KAP-GFP in the *ppr2* mutant (*fla3 ppr2* KAP-GFP) cell. This movie was taken at 20 fps and plays in real time. Scale bar 5 µm.
https://elifesciences.org/articles/82901/figures#video6

a faster disassembly rate. We therefore tested how flagellar $Ca^{2+}$ affects flagellar disassembly using the temperature-sensitive *fla10* mutant cells (*Huang et al., 1977*). The *fla10* mutant cells gradually reduce their flagellar length at the restrictive temperature (*Huang et al., 1977*; *Marshall et al., 2005*). The *ppr2* mutation had no clear effect on disassembly rates, and EGTA-treated cells showed faster disassembly kinetics than untreated cells in both control and *ppr2* mutant cells (*Figure 5B*), rather than the slower kinetics that would be predicted if flagellar $Ca^{2+}$ normally promoted disassembly. We also tested putative mutations in flagella-localized calcium pumps and found no effect on flagellar length (*Figure 5—figure supplement 1*). Overall, our results suggest that flagellar $Ca^{2+}$ does not inhibit IFT injection or flagellar assembly or promote flagellar disassembly.

## Flagellar $Ca^{2+}$ does not change IFT injection even at short timescales

Our analyses suggested that flagellar $Ca^{2+}$ helps flagellar assembly and maintenance rather than flagellar disassembly and does not inhibit IFT injection. However, this conclusion is based on separate measurements of $Ca^{2+}$ and IFT carried out under identical conditions but not in the same cells, and is based on comparison of population averages as a function of length, leaving open the possibility that there could be correlations between $Ca^{2+}$ influx and IFT injection over short periods of time. Such a short time scale correlation might be important given that calcium shows spiking dynamics that are more regularly spaced than predicted by a Poisson distribution. These spikes could play a role in transiently delaying or shifting the time at which IFT injections occur, leading to a more regular or uniform IFT time series. To ask how flagellar $Ca^{2+}$ might contribute to flagellar length control or IFT injection over short time scales in individual flagella, we performed dual-channel live-cell imaging of $Ca^{2+}$ and IFT using DRC4-GCaMP and mScarlet-IFT54 (*Figure 6A* and *Video 9*). IFT injection measured in dual-channel imaging was negatively correlated with flagellar length, similar to our KAP-GFP analysis (*Figure 6B*). Both frequency and speed of anterograde mScarlet-IFT54 showed a similar distribution with KAP-GFP (*Figures 4D, E, 6C and D*). Likewise, GCaMP intensity measured in dual-channel imaging also showed a similar distribution as previously seen in our single-wavelength imaging (*Figure 6E*). However, there was no obvious correlation between $Ca^{2+}$ influx and IFT injection (*Figure 6F*), consistent with our results above comparing separate single-channel imaging data.

To further investigate the potential effect of $Ca^{2+}$ influx on IFT behavior on a short time scale, we asked whether a burst of $Ca^{2+}$ influx might cause

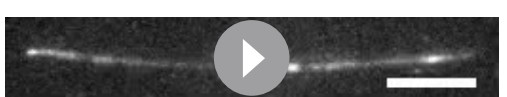

**Video 7.** Movie of KAP-GFP in 1 mM EGTA treated (*fla3* KAP-GFP) cell. This movie was taken at 20 fps and plays in real time. Scale bar 5 µm.
https://elifesciences.org/articles/82901/figures#video7

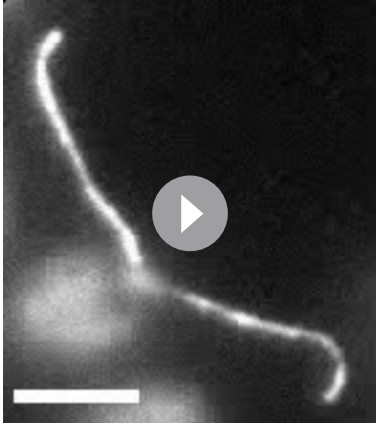

**Video 8.** Movie of KAP-GFP in 1 mM CaCl₂ treated (*fla3* KAP-GFP) cell. This movie was taken at 20 fps and plays in real time. Scale bar 5 µm.
https://elifesciences.org/articles/82901/figures#video8

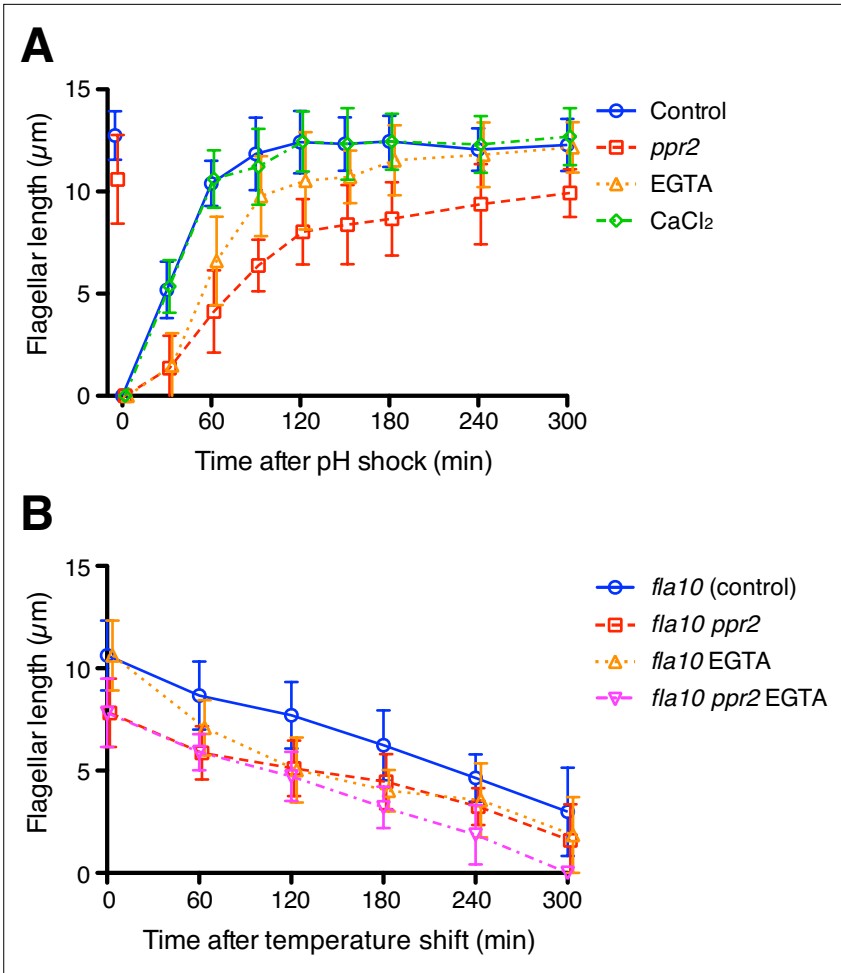

**Figure 5.** The kinetics of flagellar regeneration and disassembly. (**A**) Flagella were removed by the pH shock method. Mean flagellar length of control (wild-type CC-125, blue circles), *ppr2* mutant (red squares), 1 mM EGTA-treated (orange triangles), and 1 mM CaCl₂-treated (green diamonds) cells was plotted against time after pH shock. (**B**) The *fla10* mutant strains were transferred to the restrictive temperature (33 °C). Mean flagellar length of control (*fla10*, blue circles), *fla10 ppr2* mutant (red squares), 1 mM EGTA-treated *fla10* (orange triangles), and 1 mM EGTA-treated *fla10 ppr2* (magenta inverted triangles) cells was plotted against time after temperature shift. Twenty biflagellated cells were measured per strain and time point. Data were plotted with mean ± SD.

The online version of this article includes the following source data and figure supplement(s) for figure 5:

**Source data 1.** Raw data of flagellar length during flagellar regeneration and disassembly assays.

**Figure supplement 1.** Flagellar length of potential flagellar calcium pump mutant cells.

---

a transient change in IFT behavior. We selected kymograph regions in which there was no observed Ca²⁺ influx for at least 10 s before a Ca²⁺ burst and analyzed successive 5 s intervals (**Figure 7A**). In this analysis, we did not find a statistically significant difference in intensity or frequency of IFT injection during the time windows before or after Ca²⁺ influx (**Figure 7B, C**). However, average speeds of both anterograde and retrograde IFT appeared to increase during Ca²⁺ influx. The increase in average speed was due to the fact that paused IFT trains, which had an initial speed of zero, became re-activated during the Ca²⁺ bursts, consistent with a previous report (**Collingridge et al., 2013**; **Fort et al., 2021**). In experiments in which cells were treated with 1 mM EGTA, a slowdown of IFT trains was also observed (**Figure 4E**). Moreover, the background intensity of the mScarlet-IFT54 channel decreased after Ca²⁺ influx (**Figure 7I**). This background reduction means that IFT trains, which paused and accumulated during the prolonged absence of Ca²⁺ influx, restarted with Ca²⁺ influx. Because most of the pausing IFT trains were retrograde IFT, the intensity of the returning IFT was transiently increased after Ca²⁺ influx (**Figure 7C**). This result showed that Ca²⁺ bursts can clear out the accumulated retrograde

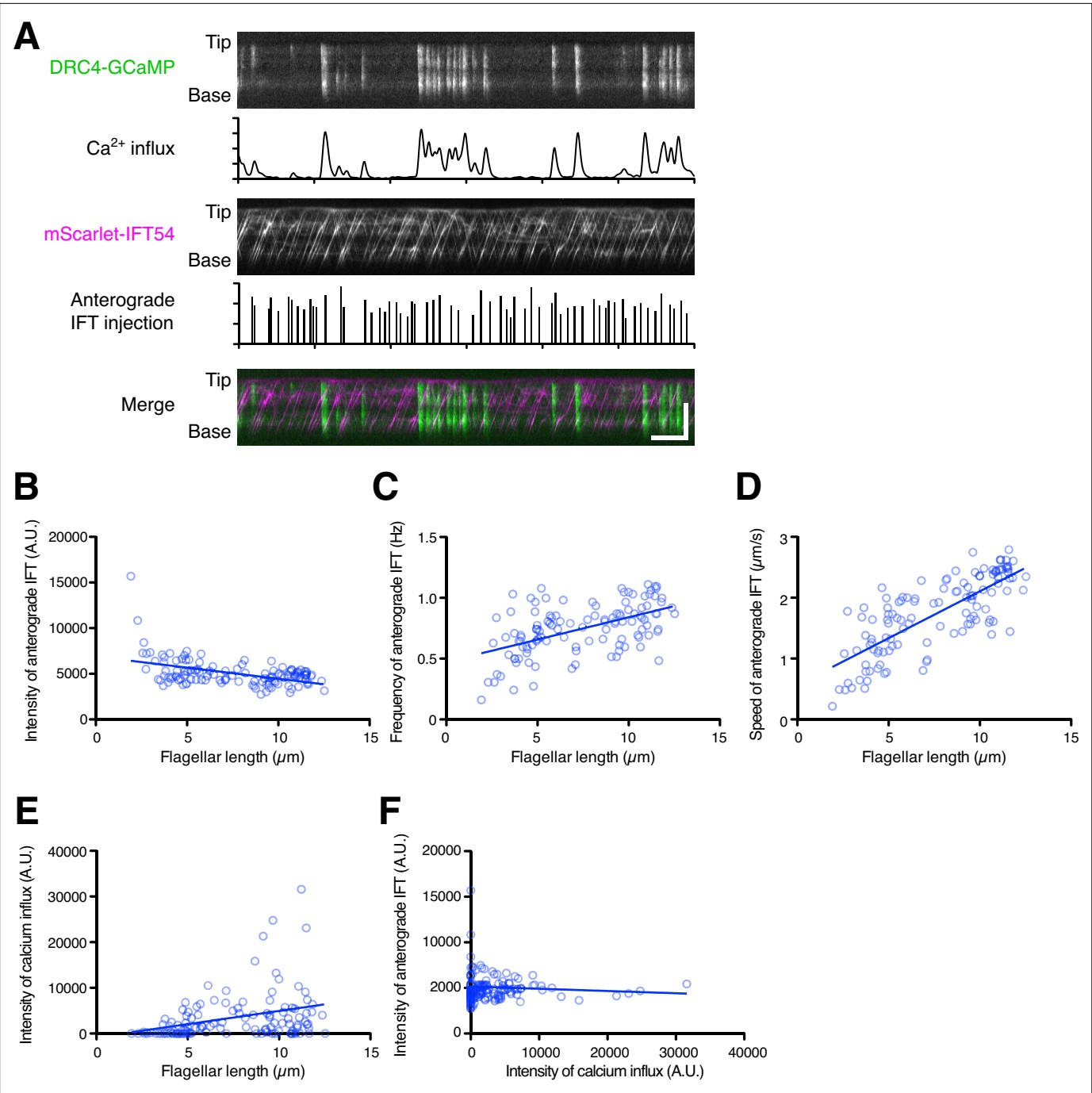

**Figure 6.** Dual-channel imaging of DRC4-GCaMP and mScarlet-IFT54. (**A**) Representative kymographs and quantified data. The DRC4-GCaMP kymograph (top) was generated from **Video 9**. Quantified intensity of $Ca^{2+}$ influx is shown at the bottom of the kymograph. The mScarlet-IFT54 kymograph (middle) was generated from **Video 9**. Quantified anterograde IFT injection is shown at the bottom of the kymograph. The merged kymograph (bottom) of DRC4-GCaMP (green) and mScarlet-IFT54 (magenta). Horizontal bar: 5 s; vertical bar: 5 μm. (**B**) The mScarlet-IFT54 intensity of each flagellum was calculated from kymographs and plotted against flagellar length (n=116, and Pearson correlation coefficient $\rho$ =−0.47). Non-linear regression is indicated by a solid line ($r^2$=0.21). (**C**) The frequency of anterograde IFT was plotted against flagellar length ($\rho$ =0.53 and $r^2$=0.28). (**D**) The velocity of anterograde IFT was plotted against flagellar length ($\rho$ =0.76 and $r^2$=0.58). (**E**) The intensity of $Ca^{2+}$ influx into flagella was calculated from kymographs and plotted against flagellar length ($\rho$ =0.33 and $r^2$=0.11). (**F**) The mean mScarlet-IFT54 intensity was plotted against the intensity of $Ca^{2+}$ influx ($\rho$ =−0.08 and $r^2$=0.007).

The online version of this article includes the following source data for figure 6:

**Source data 1.** Raw data of flagellar length and intensities of DRC4-GCaMP and mScarlet-IFT54.

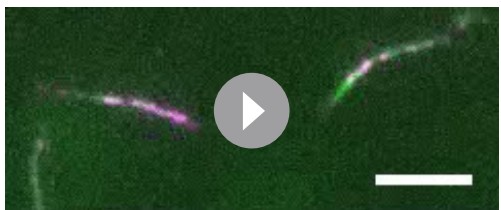

**Video 9.** Movie of DRC4-GCaMP and mScarlet-IFT54 in the *pf2 ift54* DRC4-GCaMP mScarlet-IFT54 cell. DRC4-GCaMP is shown in green, and mScarlet-IFT54 is shown in magenta. This movie was taken at 10 fps and plays in real time. Scale bar 5 μm.

https://elifesciences.org/articles/82901/figures#video9

IFT trains from flagella. Additional examples are given in *Figure 7—figure supplement 1*. Because a portion of IFT trains, after returning to the base, can be reused as new anterograde IFT trains (*Wingfield et al., 2017*), we expected that IFT injection might increase after a burst of $Ca^{2+}$ influx, due to the resulting release of paused retrograde trains. However, IFT injection did not obviously increase after the returning IFT trains reached the base of the flagellum in this time range (*Figure 7B*). Together, these results indicate that $Ca^{2+}$ influx can restart paused IFT trains and increase the amount of returning IFT trains to the base of flagella, but that IFT injection is not controlled by $Ca^{2+}$ influx in a short time and is not triggered by a transient increase in the arrival of retrograde trains.

## Discussion

### The role of $Ca^{2+}$-dependent phosphorylation in IFT

The work of *Liang et al., 2014* has clearly shown that the IFT kinesin is regulated by CDPK1. Why then in our hands do we not observe the expected effect of calcium channel modulation and $Ca^{2+}$ depletion on length? For one thing, it has not been shown directly that this kinase requires $Ca^{2+}$ for its activity in *Chlamydomonas*. CDPK1 has also been shown to regulate IFT turnaround at the flagellar tip (*Liang et al., 2014*); however, recent studies showed that IFT can turn around at the tip without $Ca^{2+}$ (*Chien et al., 2017*; *Nievergelt et al., 2022*). Thus, $Ca^{2+}$ might not be required for the activation of CDPK1. Furthermore, it is important to note that regulation does not have to imply feedback control. We modeled the phosphorylation of kinesin as occurring in response to $Ca^{2+}$ entering through the flagellar membrane, which is consistent with the original ion-current model (*Rosenbaum, 2003*). However, it is also possible that the phosphorylation is regulated by cytoplasmic $Ca^{2+}$ fluctuations, which could be independent of the flagellum, thus constituting a separate regulatory input. In such a scenario, CDPK1 would not be part of a feedback loop because it would not be sensitive to flagellar length. This case may therefore be similar to the length-regulating kinase LF4 in *Chlamydomonas*. Null mutants in *lf4* have flagella that are approximately double the normal length, but their length is still regulated in the sense that length variation is constrained (*Bauer et al., 2021*). Since length is still regulated in a null mutant of *lf4*, the kinase cannot be part of a feedback loop essential for length control. Instead, it is likely to carry information from other signaling pathways to alter the set-point of flagellar length. We speculate that the calcium-dependent regulation of kinesin entry into flagella during IFT may provide a way for cellular states, such as cell cycle or metabolic state, to influence flagellar length, and not as a feedback control loop necessary for length regulation per se. *Liang and Pan, 2013* also showed that another calcium-dependent protein kinase, CDPK3, helps flagellar assembly at low $Ca^{2+}$ concentrations. CDPK1 and CDPK3 may have opposite regulation over length control downstream of $Ca^{2+}$. We speculate that regulation of some flagella related processes by cytoplasmic, rather than flagellar $Ca^{2+}$, may explain the long-known result that reduction in $Ca^{2+}$ leads to shortening or loss of flagella (*Lefebvre et al., 1978*) which has always stood in contradiction to the ion-current model.

### The role of flagellar $Ca^{2+}$ influx in flagellar length control

The ion-current model assumed that the $Ca^{2+}$ influx was stable. However, the amount of flagellar $Ca^{2+}$ was highly dispersed around the trend line although on average it is correlated with flagellar length. This variability mostly came from the randomness of the $Ca^{2+}$ influx events. However, the $Ca^{2+}$ influx events did not match a Poisson distribution, consistent with the idea of $Ca^{2+}$ bursting. $Ca^{2+}$ influx may itself be regulated by some other process. A previous study found that increased $Ca^{2+}$ influx is evoked in the trailing flagellum during gliding motility (*Collingridge et al., 2013*). Because our cells were also mounted on glass surfaces to image flagella, the gliding motility could also be one source of variation

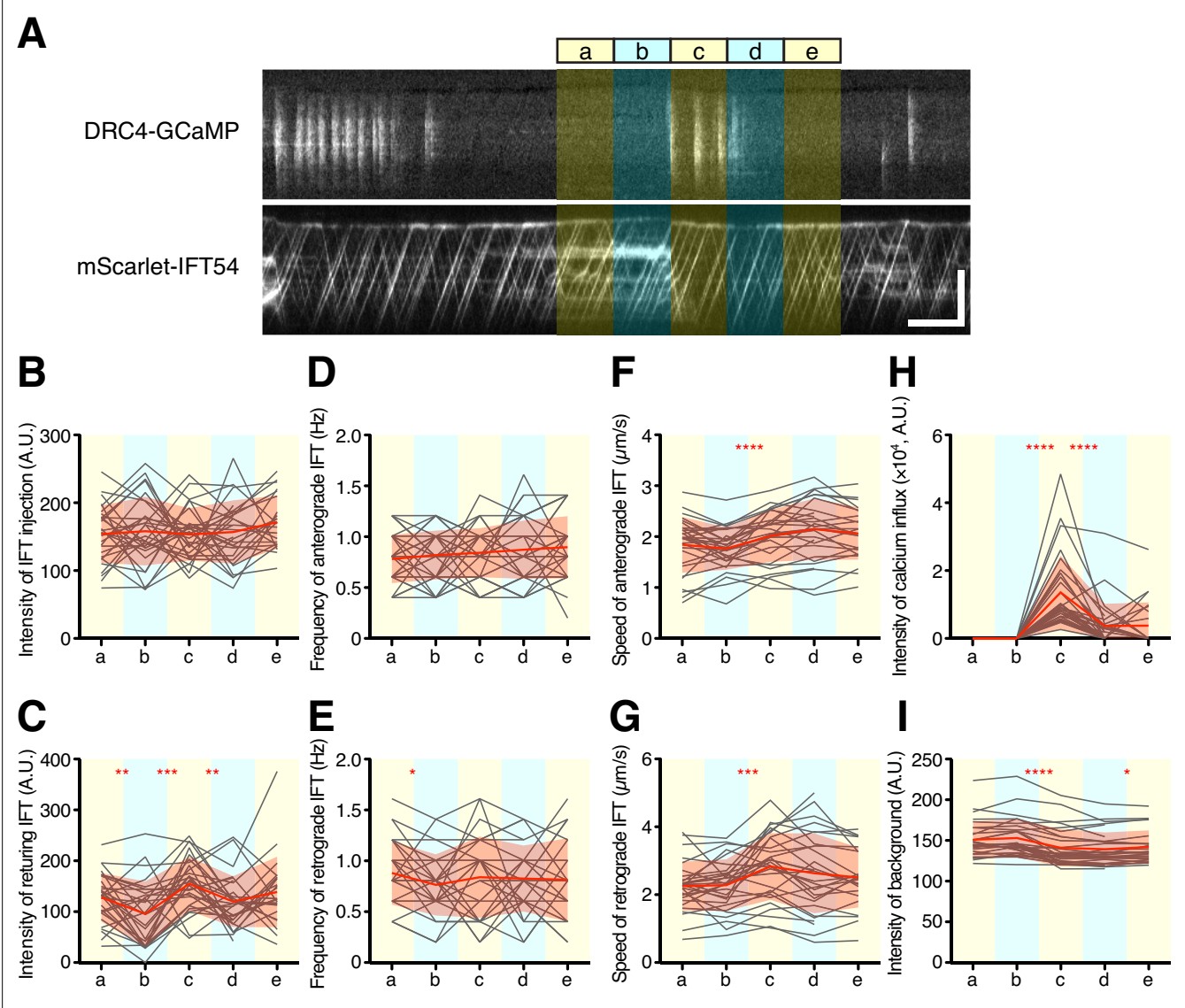

**Figure 7.** Quantification of IFT behavior after Ca²⁺ influx. (**A**) Overview of the short-time analysis of IFT behavior after a burst of Ca²⁺ influx using dual-channel imaging. Analyzed areas were colored on the representative kymographs of DRC4-GCaMP and mScarlet-IFT54. Intervals of each area are 5 s. Bursts of Ca²⁺ influx were selected where no previous Ca²⁺ influx was observed for more than 10 s before the Ca²⁺ influx. The 'c' area was set at the start of the burst of Ca²⁺ influx. The areas 'a' and 'b' were set to 10 s and 5 s before the burst of Ca²⁺ influx at the 'c' area. The areas 'd' and 'e' were set to 5 s and 10 s after the start of the Ca²⁺ influx, respectively. Horizontal bar: 5 s; vertical bar: 5 µm. (**B–G**) IFT behaviors before and after the burst of Ca²⁺ influx were plotted against each area (n=29). Gray lines show each analysis, and red lines and area show the mean and SD, respectively. (**B**) The intensity of IFT injection. (**C**) The intensity of returning IFT. (**D**) The frequency of anterograde IFT. (**E**) The frequency of retrograde IFT. (**F**) The speed of anterograde IFT. (**G**) The speed of anterograde IFT. (**H**) The sum of the Ca²⁺ influx before and after the burst of Ca²⁺ influx. (**I**) The background intensity of mScarlet-IFT54. Statistical significance between neighboring areas was determined by a paired two-tailed t test (* $p<0.05$; ** $p<0.01$; *** $p<0.001$; **** $p<0.0001$).

The online version of this article includes the following source data and figure supplement(s) for figure 7:

**Source data 1.** Raw data of IFT behavior analysis.

**Figure supplement 1.** Additional examples of stalled IFT release by Ca²⁺ influx.

in our experiments. Because we imaged cells embedded in agarose, they were prevented from gliding extensively. Previous studies have also shown that poly-lysine-coated coverslips can increase Ca²⁺ influx events in flagella (*Fort et al., 2021*). These findings suggested that a mechanical stimulus on the flagellum is one of the triggers of Ca²⁺ influx. The difference in Ca²⁺ influx between the two flagella might also be related to the phototactic behavior of *Chlamydomonas*. *Chlamydomonas* cells can change beat frequency and beat pattern of each flagellum to turn towards the light source (*Rüffer*

*and Nultsch, 1991*), and they are controlled by Ca²⁺ (*Kamiya and Witman, 1984*). This phototactic response is mediated by the signal transduction from the photoreceptor, which is located at the eyespot (*Harz and Hegemann, 1991*). Because these kinds of Ca²⁺ responses are dependent on the specific local environment in which the cell might find itself, they are unlikely to produce a stable signal, making them less useful as an indicator of flagellar length.

The flagellar regeneration kinetics of 1 mM EGTA-treated cells (*Figure 5A*) were similar to results reported in rapamycin treated *Chlamydomonas* cells. Rapamycin-treated cells also showed slower flagella regeneration kinetics, but the length of the flagellum eventually reached its original length before deflagellation (*Yuan et al., 2012*). Rapamycin inhibits the mTOR pathway (*Wullschleger et al., 2006*), and it is known that intracellular Ca²⁺ is necessary for the activation of the mTOR pathway (*Gulati et al., 2008*; *Li et al., 2016*). The slow regeneration kinetics of EGTA-treated cells might be caused by the inhibition of the cytoplasmic mTOR pathway. Other Ca²⁺ dependent processes in the cytoplasm might also potentially affect IFT, and our results cannot rule out this possibility. However, we note that the *ppr2* mutant also fails to show the effect on IFT or regeneration predicted by the ion current model.

It has been proposed that not only IFT injection, but also cargo loading onto IFT particles, is regulated as a function of flagellar length (*Craft et al., 2015*). Quantitative studies show that some or all of this apparent length-dependent cargo loading is a result of the fact that in longer flagella, IFT trains are shorter and thus have fewer tubulin binding sites (*Wemmer et al., 2020*). Nevertheless, the question arises whether flagellar Ca²⁺ could play a role in regulating tubulin binding to the IFT particles. If this was the case, Ca²⁺ would have to inhibit cargo binding, in order for cargo binding to decrease at longer lengths as has been reported. Reduction of flagellar Ca²⁺ should thus lead to increased cargo binding, causing flagella to grow faster and/or to longer lengths. Our measurements of flagellar growth rate (*Figure 5A*) show that in fact the opposite is true.

Taken together, our experimental test of the ion-current model for flagella length control appears to rule out an ion channel model based on Ca²⁺ influx into flagella. It is formally possible that some other ion might serve as a length sensor, but thus far Ca²⁺ is the only candidate ion for which a plausible mechanistic link to IFT regulation has been reported.

We previously provided experimental evidence against two other potential length control models - an 'initial bolus' model (*Ludington et al., 2015*) in which IFT particles are initially injected into a newly formed flagellum but then recycle without being able to leave, as well as a 'time-of-flight' model (*Ishikawa and Marshall, 2017*; *Ludington et al., 2015*) in which IFT particles contain a molecular timer that tracks the time spent inside the flagellum as a proxy for flagellar length. Here we rule out a calcium-based ion-current model for length sensing as well. Of the models we have considered, the remaining model for length-dependent IFT regulation is one based on diffusive return of the IFT kinesin (*Hendel et al., 2018*). The experiments presented here do not address this diffusion-based model, which is thus far the only model for which there is no opposing experimental evidence.

## Return of re-activated retrograde trains

When anterograde IFT trains reach the tip, they remodel into a different number of retrograde trains (*Chien et al., 2017*). Much less is known about possible remodeling of retrograde trains at the basal body in order to re-inject them. One possibility might be that some of returning IFT particles are immediately packaged into anterograde trains and re-injected back into the same flagellum (*Wingfield et al., 2017*).

We observed the re-activation of large paused retrograde trains by Ca²⁺ spikes, which has been reported to occur due to an ability of Ca²⁺ influx to break the link between IFT and membrane proteins and thus restart paused retrograde IFT trains (*Fort et al., 2021*). We note in addition that many cytoplasmic dynein cargo adaptors contain EF hand domains (*Reck-Peterson et al., 2018*), and at least in the case of CRACR2a it has been shown that increased cellular calcium can activate dynein motility (*Wang et al., 2019*). Regardless of the mechanism, the re-activation of these paused trains results in the sudden return of a large quantity of IFT particles to the base, as visualized by a strong retrograde trace. However, when we then looked at the anterograde trains entering the flagellum in a short time period after the return of these trains, no significant increase in IFT injection was observed (*Figure 7*). This observation implies that returning IFT particles are not directly re-injected. This may indicate that the remodeling/recycling of retrograde IFT trains into anterograde IFT trains simply needs more time,

or it might be because IFT injection is controlled by some other process that determines the timing of new particle injection, regardless of instantaneous retrograde return events.

## Materials and methods

### *Chlamydomonas* strains and culture condition

*Chlamydomonas reinhardtii* strains, wild-type 137 c (CC-125), the *fla3-1* mutant expressing KAP-GFP (CC-4296)(*Mueller et al., 2005*), the *fla10-1* mutant (CC-1919) (*Huang et al., 1977*), and the *pf2-4* mutant (CC-4404)(*Rupp and Porter, 2003*) were obtained from the Chlamydomonas Resource Center (University of Minnesota, St. Paul, MN). The *ppr2* mutant strain was provided by Kenjiro Yoshimura (*Fujiu et al., 2009*; *Matsuda et al., 1998*). The *ift54-2* mutant strain expressing mScarlet-IFT54 was provided by Karl Lechtreck (*Fort et al., 2021*; *Wingfield et al., 2017*). The generation of *pf2-1* DRC4-GCaMP has been described (*Nievergelt et al., 2022*). The presence of mutant alleles was confirmed by PCR. *pf2-1* DRC4-GCaMP was crossed with the *pf2-4* mutant. We crossed the *pf2-4* DRC4-GCaMP strain with the *ppr2* strain to generate the *ppr2 pf2-4* double mutant strain expressing DRC4-GCaMP. We generated the *ppr2* mutant expressing KAP-GFP by crossing the *ppr2* mutant with the KAP-GFP strain. For dual-channel imaging of Ca$^{2+}$ and IFT, we crossed the *pf2-4* DRC4-GCaMP strain with *ift54-2* mScarlet-IFT54 strain to generate the *pf2-4 ift54-2* double mutant strain expressing both DRC4-GCaMP and mScarlet-IFT54. For flagellar disassembly assays, the *fla10-1* mutant strain was crossed with the *ppr2* strain to generate the *fla10-1 ppr2* double mutant strain. The potential flagellar calcium pump mutant strains were obtained from *Chlamydomonas* Library Project (CLiP)(*Li et al., 2019*). *Chlamydomonas* cells were grown in liquid Tris-acetate-phosphate (TAP) medium media (20 mM Tris HCl, 3.5 mM NH$_4$Cl, 0.2 mM MgSO$_4$, 0.17 mM CaCl$_2$, 1 mM K$_3$PO$_4$, and 1000×diluted Hutner's trace elements, titrated to pH 7.0 with glacial acetic acid) (*Harris et al., 2009*) at room temperature with constant aeration in light.

### Flagellar regeneration and disassembly assays

Flagellar regeneration assays were performed in the same manner as previously described (*Ishikawa and Marshall, 2017*). Briefly, *Chlamydomonas* liquid cultures were adjusted to pH 4.5 by adding 0.5 M acetic acid and incubated for 1 min to amputate flagella, and then the pH was returned to pH 7.0 with 0.5 M KOH. The pH shocked cells were pelleted by centrifugation and resuspended with fresh TAP media with or without 1 mM EGTA (free Ca$^{2+}$ is 149.3 nM and 170 µM, respectively). For the flagellar disassembly assay, we used the *fla10* temperature-sensitive kinesin mutant. The *fla10* mutant and *fla10 ppr2* double mutant cells were cultured in TAP media at room temperature for 1 day and then 1 mM EGTA was added, after which the cultures were transferred to a 33 °C incubator with continuous illumination. Cells were fixed with 2.5% glutaraldehyde before and at time points after pH shock or temperature shift. Fixed cells were imaged by differential interference contrast (DIC) microscopy using an inverted microscope (AxioVert 200 M, Zeiss, Jena, Germany) with an air objective (Plan Apo 40 x/ NA 0.75, Zeiss) and a CCD camera (Axiocam MRm, Zeiss). The lengths of 20 biflagellated cells at each time point and each sample were measured using ImageJ (NIH, Bethesda, MD). We often observed the *fla10* mutant and *fla10 ppr2* double mutant cells with a single flagellum after the temperature shift but measured only the cells with two flagella. We did not observe any *fla10 ppr2* double mutant cells with two flagella at 300 min after the temperature shift.

### Live-cell imaging

All live-cell imaging was performed during the regeneration of flagella. *Chlamydomonas* flagella were amputated by the pH shock method as described above. #1 (22 x 22 mm) coverslips were coated with 0.1% poly-L-lysine solution (Sigma, St. Louis, MO) for 10 min and washed three times with deionized water. A culture of flagella-regenerating cells was placed on a poly-L-lysine-coated coverslip for 3 min and removed most of the media. Then, the coverslip was mounted on glass slides with 2% agarose/TAP with or without 1 mM EGTA and sealed with VALAP (1:1:1 Vaseline, lanolin, and paraffin). Agarose embedding prevents cells from gliding. Images were acquired on an inverted microscope (Eclipse Ti-E, Nikon, Tokyo, Japan) configured for TIRF microscopy using an oil objective (Apo TIRF 100 ×/NA 1.49; Nikon), MLC400 laser combiner with solid-state lasers of 405, 488, 561, and 647 nm (Keysight, Santa Rosa, CA) and an electron-multiplying CCD (EM-CCD; iXon Ultra 897, Andor, Belfast,

UK) camera controlled with Nikon Elements Software (v5.20.00 b1423). Image sequences were continuously acquired using triggered acquisition using the 488 nm and/or 561 nm lasers (4.0 mW and 1.6 mW, respectively) and single band-pass (ET525/50 m and ET600/60m, Chroma, Bellows Falls, VT) or dual band-pass emission filters (59012m, Chroma) with a camera exposure of 50 ms and an EM gain of 300 used for both channels.

## Quantifying axonemal DRC4-GCaMP

To quantify the GCaMP reactivity, axonemes were isolated from the *pf2-4* DRC4-GCaMP or *pf2-4* DRC4-mCherry-GCaMP strains. Flagellar isolation and demembranation were performed as previously described (*Wakabayashi and Kamiya, 2015*). For Ca$^{2+}$ titrations, GCaMP axonemes were washed and resuspended in 30mM MOPS, pH 7.2, 100mM KCl with either 10mM EGTA (zero free Ca$^{2+}$) or 10mM CaEGTA (39μM free Ca$^{2+}$) from the calcium calibration buffer kit (Thermo Fisher, Waltham, MA) and then observed on the TIRF microscope (Eclipse Ti-E, Nikon). GCaMP intensity was measured using ImageJ. Each axoneme was traced by hand using the Segmented line tool with a 7-pixel line width, analyzed by the Plot Profile function, and then average intensity per pixel was calculated. At least 13 axonemes were analyzed for each calcium concentration.

## Imaging analysis

For GCaMP intensity analysis, kymographs were generated from movies of DRC4-GCaMP using ImageJ. GCaMP kymographs were analyzed in MATLAB (MathWorks, Natick, MA). The kymographs were smoothed using a 2D Gaussian filter and photobleach-corrected. The peak of Ca$^{2+}$ influx was detected using the findpeaks function on MATLAB. The intensity of Ca$^{2+}$ influx was calculated as the sum of the peak area of Ca$^{2+}$ influx divided by imaging time.

For quantitative analysis of IFT injection and behavior, kymographs were generated from movies of KAP-GFP or mScarlet-IFT54 using Kymograph Clear (*Mangeol et al., 2016*). IFT trajectories were detected using KymoButler, a deep learning-based software (*Jakobs et al., 2019*), and extracted as coordinates. IFT kymographs and coordinates of IFT trajectories were analyzed in MATLAB. Whenever separated IFT trajectories were detected on the same track, they were connected, and any overlaps of IFT trajectories were removed. The IFT injection intensity was calculated as the total intensity of anterograde IFT trajectories per unit length and unit time. IFT velocity and frequency were calculated from extracted IFT trajectories. The intensities of IFT injection and returning IFT were calculated as the sum of the average intensity of injecting or returning IFT trajectories during the image acquisition period.

For analysis of dual-channel imaging, dual-channel movies were converted to kymographs for each channel. The mScarlet channel showed slight bleed-through to the GCaMP channel because we used the dual band-pass emission filter for imaging. To remove the weak bleed-through of mScarlet-IFT54 signals from DRC4-GCaMP kymographs, 20% intensity of the mScaret-IFT54 kymograph was subtracted from the GCaMP kymograph. The intensity of Ca$^{2+}$ influx was calculated in the same manner as described above. To investigate the short-term effect of Ca$^{2+}$ influx on IFT, we selected the kymographs without Ca$^{2+}$ influx for more than 10 s before bundles of Ca$^{2+}$ influx that had more than 2000 intensities. We calculated IFT injection, velocity, and frequency for 10 s and 15 s each before and after the start of the Ca$^{2+}$ influx in these kymographs. The intensities of IFT injection and returning IFT were calculated as the sum of the intensity of IFT trajectories that started from or returned to the base within the time range. We also calculated the background intensity from the kymographs which removed both anterograde and retrograde IFT trajectories. All calculations were performed using custom-written routines in MATLAB.

For statistical analyses and graph generation, we used GraphPad Prism (GraphPad, San Diego, CA). The statistical significance of Ca$^{2+}$ influx between the control and the *ppr2* mutant was tested using an unpaired t test. Statistical significances of IFT behaviors were tested using a paired t test.

## Computational model

We constructed a computational model of flagellar length control based on the general concept proposed by *Rosenbaum, 2003* and based on the biochemical work of Liang et al on CDPK1 mediated phosphorylated of kinesin-2 (*Liang et al., 2014*). This model is based on the following assumptions. First, as first proposed by *Rosenbaum, 2003* based on the measurements of *Beck and Uhl,*

*1994*, we assume that the concentration of calcium at the base is on average proportional to the length of the flagellum.

$$Ca^{2+} = \alpha L$$

where α is a proportionality constant.

Second, we assume that CDPK1 activity depends on the instantaneous calcium concentration according to a saturable binding relation with some dissociation constant that is an adjustable parameter of the model. In effect, this assumption means that the kinetics of allosteric regulation of CDPK1 activity by calcium occur on a much faster time scale than the other events of our model. This yields the relation:

$$F_{CDPK} = \frac{\alpha L}{\alpha L + K_D}$$

where $F_{CDPK}$ is the fraction of enzyme that is active, and $K_D$ is the dissociation constant of calcium binding by CDPK1.

Third, we assume that phosphorylation of kinesin by CDPK1, as well as subsequent dephosphorylation of kinesin, both take place on a fast timescale relative to the timescale of flagellar length change, such that the fraction of active kinesin is at a quasi-state in the model. Given this assumption, we represent the fraction of phosphorylated kinesin as a function of the relative activity of CDPK1 according to a Michaelis-Menten equation:

$$K = 1 - \frac{F}{F + K_M}$$

where $K_M$ is the Michaelis constant for the interaction of the enzyme CDPK1 with the kinesin substrate.

Fourth, based on the results of *Liang et al., 2014*, we assume that the number of IFT particles entering the flagellum per unit time is proportional to the fraction of non-phosphorylated kinesin-2 proteins. Fifth, we assume that the occupancy of IFT particles for tubulin is proportional to the quantity of tubulin available in the cytoplasm, which is equivalent to saying that binding is not saturated. This was the assumption used in previous models (*Marshall et al., 2005*) and is consistent with observations on the fractional occupancy of tubulin during flagellar assembly (*Craft et al., 2015*; *Wemmer et al., 2020*). We thus represent the available tubulin pool as (P-2L) where P is the total precursor (tubulin) pool and L is the length of each of the two flagella. With these assumptions, we can describe flagellar length dynamics as follows:

$$\frac{dL}{dt} = A * K * (P - 2L) - D$$

where A is a proportionality constant describing the total pool of kinesins, such that AK is the pool of kinesins currently active, as well as the incremental increase in length that occurs when a kinesin delivers cargo to the tip. D is the rate of disassembly, which is assumed to be constant based on prior measurements (*Marshall et al., 2005*).

The net growth rate is used to update the length using the Euler method with a time-step of 0.02 s. Output values are stored every 50 timesteps (1 second of simulation). Source codes of the model and data analyses used in this work (*Ishikawa and Marshall, 2022*) can be found here: https://github.com/ishikawaUCSF/IonCurrentModel, (copy archived at swh:1:rev:c916c6101e469bab1b1f0b6fa2a79bc56266c390).

## Acknowledgements

The authors thank current and past members of the Marshall laboratory as well as Joel Rosenbaum and Xin Xiang for helpful discussions; DeLaine Larsen, Kari Herrington, SoYeon Kim and the Nikon Imaging Center at UCSF for microscopy resources and assistance; and Karl Lechtreck (University of Georgia) and Kenjiro Yoshimura (Shibaura Institute of Technology) for generously sharing *Chlamydomonas* strains. This work was supported by NIH grants R35GM130327 (WFM) and R01GM130908 (MD).

# Additional information

## Funding

| Funder | Grant reference number | Author |
| --- | --- | --- |
| National Institutes of Health | R35GM130327 | Wallace F Marshall |
| National Institutes of Health | R01GM130908 | Markus Delling |

The funders had no role in study design, data collection and interpretation, or the decision to submit the work for publication.

## Author contributions

Hiroaki Ishikawa, Conceptualization, Software, Validation, Investigation, Visualization, Methodology, Writing - original draft, Writing - review and editing; Jeremy Moore, Software; Dennis R Diener, Markus Delling, Conceptualization, Resources, Writing - review and editing; Wallace F Marshall, Conceptualization, Software, Funding acquisition, Writing - original draft, Project administration, Writing - review and editing

## Author ORCIDs

Hiroaki Ishikawa ⓘ http://orcid.org/0000-0003-3984-3657
Markus Delling ⓘ http://orcid.org/0000-0001-9556-2097
Wallace F Marshall ⓘ http://orcid.org/0000-0002-8467-5763

## Decision letter and Author response

Decision letter https://doi.org/10.7554/eLife.82901.sa1
Author response https://doi.org/10.7554/eLife.82901.sa2

# Additional files

## Supplementary files

• MDAR checklist

## Data availability

Modelling code has been uploaded to Github (copy archived at swh:1:rev:c916c6101e469bab1b1f0b-6fa2a79bc56266c390) and a link to the repository is provided in Materials and Methods.

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
