## [Editor Report]

This paper is valuable and of interest to scientists studying primary cilia/flagellar formation and regulation. It addresses how ciliary/flagellar length is controlled and whether calcium negatively regulates Intraflagellar transport (IFT) injection. The study convincingly demonstrates that calcium influx correlates with flagellar length, but calcium does not appear to work as a negative regulator of IFT injection, which challenges a previous model. The models and methods are generally sound.

---

## [Decision Letter]

**Decision letter after peer review:**

Thank you for submitting your article "Testing the ion-current model for flagellar length sensing and IFT regulation" for consideration by *eLife*. Your article has been reviewed by 3 peer reviewers, and the evaluation has been overseen by a Reviewing Editor and Piali Sengupta as the Senior Editor. The reviewers have opted to remain anonymous.

Essential revisions:

1) The authors only examined the impact of reducing ciliary calcium influx. To further support the conclusions, it is recommended that the authors examine IFT injection in a condition where ciliary calcium level is increased. Using a calcium ionophore may not be a good choice as it may change the global cellular calcium level. One approach to consider is to use mutants of a calcium pump present in cilia.

2) The authors conclude that calcium does not act as a negative regulator of IFT injection. However, close examination of the calcium flux in Figure 3B and IFT injection in Figure 4B of cilia less than 6 microns, suggests that calcium influx is higher in the ppr2 mutant than in control cilia and IFT injection is reduced in the mutant compared to the control. Thus, this analysis shows the opposite result of the authors' conclusion, and is supporting the previous model. These data should therefore be re-analysed focusing on the early stages of ciliary assembly. In addition, the graphs in Figure 4b, c, d are very hard to read and should be edited for clarity.

3) The conclusion on line 272-273 needs more evidence. The authors showed that addition of 1 mM CaCl2 does not change ciliary assembly, and used this to argue against the ion-current model. The addition of calcium extracellularly may not alter intracellular/intraciliary calcium level given that cells have robust systems to control calcium homeostasis. To support the authors' conclusion, one should measure the changes of calcium level in the cell/cilia under these conditions; alternatively, authors should temper their conclusion.

4) In figure 7A the authors show an example kymogram to illustrate the time course used to measure rapid changes in response to changes in ca^2+^. Prior to each DRC4-GCaMP cluster of pulses one can observe stalled mScarlet-IFT54 which is then cleared with the pulse of ca^2+^. Is this relationship observed in the other data-sets used in this figure or is this an artifact of this particular image? Earlier in the paper the frequency of ca^2+^ appeared random in relation to length. Does it instead relate to stalled IFT trains in the leading or following flagella during gliding? How were the chlamy blocked from gliding during imaging? It is not apparent from the methods.

5) In figure 4B the authors show "the mean KAP-GFP intensity of each flagellum" as a measure of total levels of IFT in the cilia at one time. In the main text this is described as "injection intensity." While total levels of IFT in the cilia are certainly influenced by injection rates it is also influenced by other factors such as IFT retrograde trafficking rates, turn around at the tip and the number of stalled trains. Were any of these other factors changed in their conditions? The authors should discuss what exactly they are measuring when they refer to injections in this paper, especially to make the paper more accessible to a broader audience. Furthermore, the authors may want to cite and discuss the Nievergelt et al. 2022 paper from Gaia Pigino's lab, which is listed in the references but not cited in the text.

*Reviewer #1 (Recommendations for the authors):*

1. Figure 1 B. It has been reported by Liang et al. (Dev Cell 2014) that phosphorylation of FLA8 dissociates FLA8 from IFT complex. Thus, it is unknown whether kinesin-2 is inactivated. I suggest that active/inactive kinesin-2 is changed to active/inactive IFT.

2. Using a condition where intraciliary calcium can be increased to analyze IFT injection, which is critical for drawing a conclusion on calcium regulation of IFT injection.

3. For Figure 3B and 4B, I suggest the authors do another mathematical analysis but only include the data for cilia shorter than about 6 micron.

*Reviewer #3 (Recommendations for the authors):*

I would like to start by thanking the authors for the work they have done.

1) In figure 4B the authors show "the mean KAP-GFP intensity of each flagellum" a measure of total levels of IFT in the cilia at one time. In the main text this is described as "injection intensity." While total levels of IFT in the cilia are certainly influenced by injection rates it is also influenced by other factors such as IFT retrograde trafficking rates, turn around at the tip and the number of stalled trains. Were any of these other factors changed in their conditions? The authors should discuss what exactly they are measuring when they refer to injections in this paper, especially to make the paper more accessible to a broader audience.

2) In figure 7A the authors show an example kymogram to illustrate the time course used to measure rapid changes in response to changes in ca^2+^. Prior to each DRC4-GCaMP cluster of pulses one can observe stalled mScarlet-IFT54 which is then cleared with the pulse of ca^2+^. Is this relationship observed in the other data-sets used in this figure or is this an artifact of this particular image? Earlier in the paper the frequency of ca^2+^ appeared random in relation to length. Does it instead relate to stalled IFT trains in the leading or following flagella during gliding? How were the chlamy blocked from gliding during imaging? I could not find it in the methods.

3) The graphs in Figure 4 b,c,d are incredibly hard to read. Color changes or redesign would be much appreciated. Perhaps turn the individual data point opacity down so the trend lines are more visible?

---

## [Author Response]

Essential revisions:1) The authors only examined the impact of reducing ciliary calcium influx. To further support the conclusions, it is recommended that the authors examine IFT injection in a condition where ciliary calcium level is increased. Using a calcium ionophore may not be a good choice as it may change the global cellular calcium level. One approach to consider is to use mutants of a calcium pump present in cilia.

We thank the reviewers for this suggestion. The calcium current model would predict that if a calcium pump mutant failed to export calcium, the increased calcium building up inside the flagellum should lead to decreased IFT entry and a shorter flagellar length. We found at least two calcium pumps in the published *Chlamydomonas* flagella proteome (Pazour et al., 2005) and ordered several mutant strains from *Chlamydomonas* Library Project (CLiP) which are annotated as affecting these pumps. We measured the flagellar length of these potential calcium pump mutant strains, but none showed a statistically significant difference in length relative to control cells. We have now included this data as Figure 5 —figure supplement 1. Because no length change was observed, we did not perform the extremely time consuming process of constructing strains that contain these mutations along with DRC4-GCaMP and KAP-GFP.

As an alternative strategy to get at this reviewer's suggestion, we measured DRC4-GCaMP and KAP-GFP intensity in 1 mM CaCl2 treated flagella and found that CaCl2 treatment increases both the flagellar calcium level (Figure 3) and IFT injection (Figure 4). This increase in IFT injection is the opposite of what the calcium current model predicts.

Based on these results, we think the calcium pump experiment is not necessary because of the following reasons. 1. These calcium pump mutants might not increase the flagellar calcium level. 2. Even if the flagellar calcium was increased in these mutants, it does not affect the flagellar length and thus our conclusions would still hold. 3. These mutant strains might still have functional calcium pumps since the existing data on calcium pumps in flagella is likely to be incomplete. 4. The CaCl2 experiment clearly increased the flagellar calcium level inside flagella, directly addressing the point that the reviewer is getting at.

2) The authors conclude that calcium does not act as a negative regulator of IFT injection. However, close examination of the calcium flux in Figure 3B and IFT injection in Figure 4B of cilia less than 6 microns, suggests that calcium influx is higher in the ppr2 mutant than in control cilia and IFT injection is reduced in the mutant compared to the control. Thus, this analysis shows the opposite result of the authors' conclusion, and is supporting the previous model. These data should therefore be re-analysed focusing on the early stages of ciliary assembly. In addition, the graphs in Figure 4b, c, d are very hard to read and should be edited for clarity.

Thank you for pointing this out, but this result of apparently low calcium influx in shorter flagella in control cells is due to bias from technical issues: it is relatively difficult to image shorter flagella in our TIRF imaging setup, because shorter flagella have less flagellar surface area to attach the coverslip. The more motile the flagella are, the more likely are the cells to detach when their flagella are short, because the bending force of the flagella is strong enough to pull them away from their small area of adhesion. This effect is much stronger in control cells than in either the ppr2 mutants or EGTA treated cells, whose flagella are less motile. This led to a reduced number of cells examined with flagella shorter than 6 μm (17 versus 34 for control and ppr2 cells, respectively). To overcome the difficulties and biased result, we observed more flagella in control cells. The new data was integrated with previous data and shown in Figure 3. The new result shows that calcium influx in control cells is indeed higher than in the ppr2 mutant cells. So, our result is remains consistent with our conclusion, and we believe that we do not have to analyze the shorter flagella separately.

Regarding the clarify of the graphs, we have updated the graphs in Figures 3 and 4 by reducing the size and opacity of each point to increase the visibility of the trend lines.

3) The conclusion on line 272-273 needs more evidence. The authors showed that addition of 1 mM CaCl2 does not change ciliary assembly, and used this to argue against the ion-current model. The addition of calcium extracellularly may not alter intracellular/intraciliary calcium level given that cells have robust systems to control calcium homeostasis. To support the authors' conclusion, one should measure the changes of calcium level in the cell/cilia under these conditions; alternatively, authors should temper their conclusion.

This is an excellent point. We have now quantitatively measured the GCaMP intensity inside flagella when 1 mM CaCl2 was added to the media. The data has now been integrated with previous data and shown in Figure 3. It is apparent in this graph that 1 mM CaCl2 treatment produces a statistically significant increase in flagellar calcium levels. We also measured KAP-GFP intensity in CaCl2 treated flagella, and found that CaCl2 addition increased IFT injection (Figure 4). We believe that these new results strengthen our conclusions, and we thank the reviewer for encouraging us to do these experiments.

4) In figure 7A the authors show an example kymogram to illustrate the time course used to measure rapid changes in response to changes in ca^2+^. Prior to each DRC4-GCaMP cluster of pulses one can observe stalled mScarlet-IFT54 which is then cleared with the pulse of ca^2+^. Is this relationship observed in the other data-sets used in this figure or is this an artifact of this particular image? Earlier in the paper the frequency of ca^2+^ appeared random in relation to length. Does it instead relate to stalled IFT trains in the leading or following flagella during gliding? How were the chlamy blocked from gliding during imaging? It is not apparent from the methods.

We have observed the clearing of stalled trains occurring at the same time as ca^2+^ influx in many cells. Fort et al. (2021), whom we cite, also showed similar results, which makes us confident that it is a general result. Nevertheless, we have now added kymographs as a supplemental figure (Figure 7 —figure supplement 1).

It is an interesting idea that stalled trains might influence the calcium spike frequency. Glen Wheeler’s group found that gliding motility induces calcium influx in trailing flagella (Collingridge et al., 2013; Fort et al., 2021), and we also observed calcium influx in trailing flagella when cells first start gliding. However, because calcium influx was observed in the absence of gliding motility, other stimuli or signals can also induce calcium influx. In any case, we found that the frequency of calcium influx was not correlated with the flagellar length. We have now added discussed about calcium influx to the Discussion section.

We embedded *Chlamydomonas* cells in an agarose gel to prevent the moving of the cell body, and found that this form of mounting greatly reduced the number of cells that showed gliding. We confirmed that this embedded method did not differ in results from the non-embedded method. We have added a note about blocking gliding with agarose in the discussion when we discuss the potential effect of gliding, and we also now note this in the Methods section.

5) In figure 4B the authors show "the mean KAP-GFP intensity of each flagellum" as a measure of total levels of IFT in the cilia at one time. In the main text this is described as "injection intensity." While total levels of IFT in the cilia are certainly influenced by injection rates it is also influenced by other factors such as IFT retrograde trafficking rates, turn around at the tip and the number of stalled trains. Were any of these other factors changed in their conditions? The authors should discuss what exactly they are measuring when they refer to injections in this paper, especially to make the paper more accessible to a broader audience. Furthermore, the authors may want to cite and discuss the Nievergelt et al. 2022 paper from Gaia Pigino's lab, which is listed in the references but not cited in the text.

We thank the reviewers for this comment, which allowed us to see that our description of our method was not clear. In fact, what we did was to extract the anterograde IFT trajectories from the kymograph, and then use only those pixels for calculating IFT injection intensity. So, we are not measuring total flagellar IFT intensity, just the intensity found in the anterograde tracks. The intensity of anterograde IFT trajectories can reflect only the IFT injection, not retrograde IFT, etc.

We also thank the reviewer for noticing our accidental failure ot cite Nievergelt et al. (2022) in the Discussion section, which we had intended to do as evidenced by our inclusion of the citation in the references. We have now added a discussion of this paper.

Reviewer #1 (Recommendations for the authors):1. Figure 1 B. It has been reported by Liang et al. (Dev Cell 2014) that phosphorylation of FLA8 dissociates FLA8 from IFT complex. Thus, it is unknown whether kinesin-2 is inactivated. I suggest that active/inactive kinesin-2 is changed to active/inactive IFT.

Liang’s paper not only discusses dissociation of FLA8 from the IFT complex, they also mention that phosphorylation inactivates kinesin and inhibits IFT entry. Nevertheless, we have replaced the words, active and inactive, with "can bind with IFT" and "can't bind with IFT" just to avoid any potential confusion.

2. Using a condition where intraciliary calcium can be increased to analyze IFT injection, which is critical for drawing a conclusion on calcium regulation of IFT injection.

This is an excellent suggestion, given that we have shown results for calcium depletion. To this end, we have now used our fluorescent strains to measure the GCaMP intensity of 1 mM CaCl2 treated flagella and found that 1 mM CaCl2 treatment does indeed increase the calcium level in flagella. Furthermore, we now show that CaCl2 treatment increases IFT injection, again in contradiction of the prediction of the calcium ion current model that more calcium should decrease IFT injection. This new experimental data has been added as Figure 3D and Figure 4BC. These results are all consistent with our original conclusions.

3. For Figure 3B and 4B, I suggest the authors do another mathematical analysis but only include the data for cilia shorter than about 6 micron.

We address this point in the reviewer's general comments above. In brief, there were very few control cells in the short length range due to increased flagellar motility in controls. Prompted by the reviewer suggestion, we analyzed additional control cells with short flagella and found that our conclusion holds that control cells have higher calcium influx even for short flagella.

Reviewer #3 (Recommendations for the authors):I would like to start by thanking the authors for the work they have done.1) In figure 4B the authors show "the mean KAP-GFP intensity of each flagellum" a measure of total levels of IFT in the cilia at one time. In the main text this is described as "injection intensity." While total levels of IFT in the cilia are certainly influenced by injection rates it is also influenced by other factors such as IFT retrograde trafficking rates, turn around at the tip and the number of stalled trains. Were any of these other factors changed in their conditions? The authors should discuss what exactly they are measuring when they refer to injections in this paper, especially to make the paper more accessible to a broader audience.

We thank the reviewers for this comment, which allowed us to see that our description of our method was not clear. In fact, what we did was to extract the anterograde IFT trajectories from the kymograph, and then use only those pixels for calculating IFT injection intensity. So, we are not measuring total flagellar IFT intensity, just the intensity found in the anterograde tracks. The intensity of anterograde IFT trajectories can reflect only the IFT injection, not retrograde IFT, stalled trains, etc.

We have now defined injection in both the abstract and introduction, which we agree is important to do.

2) In figure 7A the authors show an example kymogram to illustrate the time course used to measure rapid changes in response to changes in ca^2+^. Prior to each DRC4-GCaMP cluster of pulses one can observe stalled mScarlet-IFT54 which is then cleared with the pulse of ca^2+^. Is this relationship observed in the other data-sets used in this figure or is this an artifact of this particular image? Earlier in the paper the frequency of ca^2+^ appeared random in relation to length. Does it instead relate to stalled IFT trains in the leading or following flagella during gliding? How were the chlamy blocked from gliding during imaging? I could not find it in the methods.

We have observed the clearing of stalled trains occurring at the same time as ca^2+^ influx in many cells. Fort et al. (2021), whom we cite, also showed similar results, which makes us confident that it is a general result. Nevertheless, we have now added kymographs as a supplemental figure (Figure 7 —figure supplement 1).

It is an interesting idea that stalled trains might influence the calcium spike frequency. Glen Wheeler’s group found that gliding motility induces calcium influx in trailing flagella (Collingridge et al., 2013; Fort et al., 2021), and we also observed calcium influx in trailing flagella when cells first start gliding. However, because calcium influx was observed in the absence of gliding motility, other stimuli or signals can also induce calcium influx. In any case, we found that the frequency of calcium influx was not correlated with the flagellar length. We have now added discussed about calcium influx to the Discussion section.

We embedded *Chlamydomonas* cells in an agarose gel to prevent the moving of the cell body, and found that this form of mounting greatly reduced the number of cells that showed gliding. We confirmed that this embedded method did not differ in results from the non-embedded method. We have added a note about blocking gliding with agarose in the discussion when we discuss the potential effect of gliding, and we also now note this in the Methods.

3) The graphs in Figure 4 b,c,d are incredibly hard to read. Color changes or redesign would be much appreciated. Perhaps turn the individual data point opacity down so the trend lines are more visible?

We have now modified the plots by reducing both the size and the opacity of the markers for individual data points. We agree that this makes the graph more readable.